# Cell type-specific regulation of ciliary transition zone assembly in vertebrates

Antonia Wiegering[1], Renate Dildrop[1], Lisa Kalfhues[1,†], André Spychala[1,†], Stefanie Kuschel[1], Johanna Maria Lier[1], Thomas Zobel[2], Stefanie Dahmen[1], Tristan Leu[1,‡], Andreas Struchtrup[1], Flora Legendre[3,4], Christine Vesque[5,6], Sylvie Schneider-Maunoury[5,6], Sophie Saunier[3,4], Ulrich Rüther[1] & Christoph Gerhardt[1,*] 

## Abstract

Ciliopathies are life-threatening human diseases caused by defective cilia. They can often be traced back to mutations of genes encoding transition zone (TZ) proteins demonstrating that the understanding of TZ organisation is of paramount importance. The TZ consists of multimeric protein modules that are subject to a stringent assembly hierarchy. Previous reports place Rpgrip1l at the top of the TZ assembly hierarchy in *Caenorhabditis elegans*. By performing quantitative immunofluorescence studies in RPGRIP1L$^{-/-}$ mouse embryos and human embryonic cells, we recognise a different situation in vertebrates in which Rpgrip1l deficiency affects TZ assembly in a cell type-specific manner. In cell types in which the loss of Rpgrip1l alone does not affect all modules, additional truncation or removal of vertebrate-specific Rpgrip1 results in an impairment of all modules. Consequently, Rpgrip1l and Rpgrip1 synergistically ensure the TZ composition in several vertebrate cell types, revealing a higher complexity of TZ assembly in vertebrates than in invertebrates.

**Keywords** cilia; ciliopathies; Nphp; Rpgrip1; Rpgrip1l
**Subject Categories** Cell Adhesion, Polarity & Cytoskeleton
**The EMBO Journal (2018) 37: e97791**

## Introduction

Cilia are tiny protrusions of nearly every human cell which function as the cell's antenna mediating numerous signalling pathways essential for development and homeostasis. Dysfunctional cilia result in severe human diseases that are commonly designated as ciliopathies (Fliegauf *et al*, 2007; Gerdes *et al*, 2009; Davis & Katsanis, 2012). Ciliopathies are often caused by mutations in genes encoding transition zone (TZ) proteins (Hildebrandt *et al*, 2011; Czarnecki & Shah, 2012). The TZ represents a compartment at the base of primary cilia at the proximal end of the axoneme controlling ciliary protein entry and exit (Betleja & Cole, 2010; Craige *et al*, 2010; Omran, 2010; Benzing & Schermer, 2011; Czarnecki & Shah, 2012). Ultrastructurally, the TZ is characterised by the presence of so-called Y-links which are detected as Y-shaped densities by transmission electron microscopy (TEM) (Gilula & Satir, 1972; Reiter *et al*, 2012). Molecularly, the TZ consists of different multiprotein complexes, the Nphp1-4-8 module, the Nphp5-6 module, the Mks/B9 module and the Inversin (Invs alias Nphp2) compartment (Sang *et al*, 2011; Czarnecki & Shah, 2012). To simplify, the Nphp1-4-8 module, the Nphp5-6 module and the Inversin compartment are summarised as the Nphp modules in this study.

Retinitis pigmentosa GTPase regulator-interacting protein 1-like (Rpgrip1l alias Ftm, Nphp8 or Mks5) is considered to be the decisive organiser of TZ protein assembly in the nematode *Caenorhabditis elegans* ensuring the proper amount of all known TZ proteins (Huang *et al*, 2011; Williams *et al*, 2011; Czarnecki & Shah, 2012; Jensen *et al*, 2015; Li *et al*, 2016). In this context, Rpgrip1l deficiency leads to an impaired length of cilia in *C. elegans* (Liu *et al*, 2011; Williams *et al*, 2011). Recently, we showed that loss of Rpgrip1l also results in cilia length alterations in mice (Vierkotten *et al*, 2007; Besse *et al*, 2011; Gerhardt *et al*, 2013, 2015). In general, alterations in cilia length are a sign of ciliary dysfunction (Cui *et al*, 2011; Dowdle *et al*, 2011; Garcia-Gonzalo *et al*, 2011; Larkins *et al*, 2011). However, the role of Rpgrip1l in TZ protein assembly in vertebrates was not yet characterised. Previous studies revealed a ciliopathy phenotype of *Rpgrip1l*$^{-/-}$ murine embryos, which die, at the latest around birth (Delous *et al*, 2007; Vierkotten *et al*, 2007; Besse *et al*, 2011; Gerhardt *et al*, 2013), whereas

1  Institute for Animal Developmental and Molecular Biology, Heinrich Heine University, Düsseldorf, Germany
2  Center for Advanced Imaging (CAi), Heinrich Heine University, Düsseldorf, Germany
3  INSERM, U983, Hôpital Necker-Enfants Malades, Paris, France
4  Sorbonne Paris Cité, Faculté de Médecine, Université Paris-Descartes, Paris, France
5  Paris-Seine (IBPS) – Developmental Biology Laboratory, Institut de Biologie, CNRS, UMR7622, INSERM U1156, Paris, France
6  Sorbonne Universités, UPMC Univ Paris 06, Paris, France
   *Corresponding author. Tel: +49-(0)211/81-12236; Fax: +49-(0)211/81-15113; E-mail: Christoph.Gerhardt@hhu.de
   †Present address: Department of Cardiovascular Physiology, Heinrich Heine University, Düsseldorf, Germany
   ‡Present address: Institute of Physiology, University of Duisburg-Essen, Essen, Germany

*Rpgrip1l*$^{+/-}$ mice are viable and healthy (Vierkotten *et al*, 2007; Besse *et al*, 2011; Gerhardt *et al*, 2013). In humans, mutations in *RPGRIP1L* cause Meckel syndrome, nephronophthisis, Joubert syndrome or COACH syndrome (Joubert syndrome with congenital hepatic fibrosis) (Arts *et al*, 2007; Delous *et al*, 2007; Wolf *et al*, 2007; Doherty *et al*, 2010). Since the phenotype of *Rpgrip1l*$^{-/-}$ mouse embryos is more severe than the phenotype of many other mouse mutants suffering from the loss of a TZ protein (Czarnecki & Shah, 2012; Norris & Grimes, 2012) and since mutations in *RPGRIP1L* lead to a far broader spectrum of human ciliopathies than mutations in most other TZ protein-encoding genes (Zaghloul & Katsanis, 2010), one may expect a high ranked position of Rpgrip1l in the vertebrate TZ assembly hierarchy comparable to that in *C. elegans*.

Remarkably, the Rpgrip1l-related protein Rpgrip1 exists exclusively in vertebrates indicating that it might regulate TZ assembly in addition to Rpgrip1l. However, mutations in Rpgrip1 lead to severe eye defects in mice and humans but do not affect the development of other organs (Dryja *et al*, 2001; Ferreira, 2005; Pawlyk *et al*, 2005; Won *et al*, 2009). *Rpgrip1*$^{nmf247/nmf247}$ mice were described to produce a truncated Rpgrip1 protein and suffer from a photoreceptor degeneration that finally leads to blindness (Won *et al*, 2009). Although these mice display exclusively an impaired eye development, Rpgrip1 is not only present in the eyes but also in other murine tissues like brain, lung, heart, liver, kidney, spleen, pancreas and testis (Lu & Ferreira, 2005).

In this study, we analysed the role of Rpgrip1l in the vertebrate TZ assembly hierarchy. Using the mouse as a vertebrate model, we show that Rpgrip1l controls the TZ amount of Nphp4 via ensuring the proper amount of Cep290 (also known as Nphp6) at the TZ. In turn, the TZ amount of Invs depends on Nphp4 and the TZ amount of Nphp1 depends on Invs. Moreover, in *Rpgrip1l*$^{-/-}$ mouse embryonic fibroblasts (MEFs) and *Rpgrip1l*$^{-/-}$ murine embryonic kidneys, the amount of all analysed Nphp proteins (Nphp1, Nphp4, Invs and Cep290) at the TZ is impaired, while the amount of the Mks/B9 module proteins is unaltered, indicating a more complex TZ assembly hierarchy in vertebrates than in invertebrates. In murine embryonic limbs, the loss of Rpgrip1l affects the TZ amount of all these proteins indicating that the role of Rpgrip1l in the TZ assembly hierarchy is cell type-specific. In the cell types in which Rpgrip1l deficiency does not impair the TZ localisation of the Mks/B9 module components, Rpgrip1l and Rpgrip1 synergistically regulate the TZ amount of the Mks/B9 module members. Our findings argue for a more complex assembly hierarchy in vertebrates than in invertebrates and hence an additional security mechanism in vertebrates to prevent the development of ciliopathies.

## Results

### In mice, Rpgrip1l ensures proper TZ protein composition in a cell type-specific manner

TZ protein composition and localisation is severely disturbed by Rpgrip1l deficiency in *C. elegans* (Huang *et al*, 2011; Williams *et al*, 2011; Jensen *et al*, 2015; Li *et al*, 2016). Consequently, it is an obvious question if Rpgrip1l also plays a decisive role in vertebrate TZ organisation. To analyse the impact of Rpgrip1l on TZ assembly in

vertebrates, we used MEFs which were obtained from *Rpgrip1l*$^{-/-}$ mouse embryos (Vierkotten *et al*, 2007). Initially, we investigated the localisation of the Nphp module proteins Nphp1, Nphp4, Invs and Cep290 in wild-type (WT) and *Rpgrip1l*$^{-/-}$ MEFs. Since we did not observe a mislocalisation of these proteins but an obvious alteration in their amount (Figs 1A–E, and EV1A–C), we determined the amount of these proteins in WT and *Rpgrip1l*$^{-/-}$ MEFs via measuring fluorescence intensities directly at the TZ. Previously, it was shown that Rpgrip1l forms a complex with Nphp1 and Nphp4 (Sang *et al*, 2011). Both Nphp1 and Nphp4 protein amounts were strongly and significantly reduced at the TZ of *Rpgrip1l*$^{-/-}$ MEFs (Fig 1A and B). To test whether exogenous expression of Rpgrip1l could rescue the amount of Nphp1 at the TZ of *Rpgrip1l*$^{-/-}$ MEFs, we transfected these MEFs with a plasmid which encodes human full-length Rpgrip1l (pRpgrip1l, the murine full-length protein cannot be stably expressed for unknown reasons). 48 h after transfection, human Rpgrip1l was located at the ciliary TZ of *Rpgrip1l*$^{-/-}$ MEFs and the amount of Nphp1 was restored (Appendix Fig S1A–C). *Rpgrip1l*$^{-/-}$ MEFs displayed a significantly diminished Invs and Cep290 amount at the TZ (Fig 1C and D) indicating that the TZ amount of all analysed Nphp proteins depends on Rpgrip1l. Using super-resolution microscopy [three-dimensional structured illumination microscopy, 3D-SIM (Gustafsson *et al*, 2008; Schermelleh *et al*, 2008)] and performing fluorescence intensity quantifications after 3D reconstruction, we confirmed the reduced amount of Cep290 at the TZ of *Rpgrip1l*$^{-/-}$ MEFs (Fig 1E, and Movies EV1 and EV2) validating the precision of our measurement procedure. To test whether the significant reduction of the analysed Nphp proteins at the TZ is based on a decreased total level of these proteins or a reduced expression of the encoding genes in *Rpgrip1l*$^{-/-}$ MEFs, we examined the overall cellular amount of Invs and Cep290 via Western blot studies and the expression of *Nphp4* via qRT–PCR. The total amount of both proteins and the expression of *Nphp4* were unaltered in *Rpgrip1l*$^{-/-}$ MEFs (Fig 1F and G) indicating that the absence of Rpgrip1l does not lead to a decrease in transcription or translation of the analysed members of the Nphp modules but affects their amount at the TZ.

Former studies revealed protein interactions between Rpgrip1l and Nphp4 as well as between Rpgrip1l and Invs as well as between Nphp4 and Nphp1 (Mollet *et al*, 2002; Arts *et al*, 2007; Delous *et al*, 2007; Mahuzier *et al*, 2012). To get a deeper insight into the assembly of the murine TZ in regard to the Nphp proteins, we inactivated Nphp4 in mice (Appendix Fig S2A–E) and analysed the amount of Rpgrip1l, Nphp1, Invs and Cep290 in *Nphp4*$^{-/-}$ MEFs. We did not observe a mislocalisation of these proteins in *Nphp4*$^{-/-}$ MEFs (Fig EV1A and C–E). In *Nphp4*$^{-/-}$ MEFs, the amount of Rpgrip1l at the TZ was unaltered (Fig 2A), while the TZ amount of Nphp1 and Invs was significantly reduced (Fig 2B and C). To prove whether exogenous expression of Nphp4 is able to rescue the amount of Nphp1 at the TZ of *Nphp4*$^{-/-}$ MEFs, we transfected these MEFs with a Nphp4-encoding plasmid (pNphp4). 48 h after transfection, re-expressed Nphp4 was present at the ciliary TZ of *Nphp4*$^{-/-}$ MEFs and the amount of Nphp1 was restored (Appendix Fig S2F–H). The amount of Cep290 at the TZ remained unaltered (Fig 2D) demonstrating that Nphp4 is essential for the localisation of Nphp1 and Invs but not Rpgrip1l and Cep290 at the vertebrate TZ. Furthermore, cilia length was not altered in the absence of Nphp4 (Fig 2E). To test whether these *in vitro* data are transferable to the *in vivo* situation, we used embryonic limb buds which are a well-established

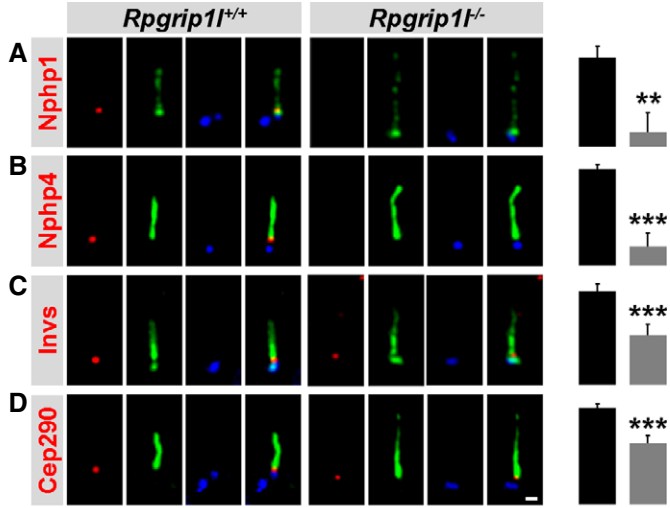

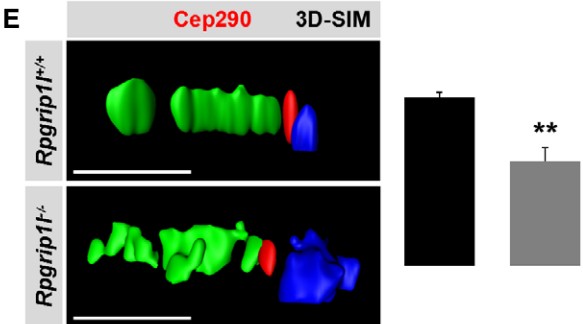

**Figure 1. Rpgrip1l regulates the proper amount of components of the Nphp modules at the ciliary TZ of MEFs.**

A–E   Immunofluorescence on MEFs which were obtained from WT and *Rpgrip1l*$^{-/-}$ embryos. (E) Z-stacks were taken by using super-resolution microscopy (3D-SIM) and reconstructed in 3D. (D and E) Scale bars (in white) represent a length of 0.5 μm (D) and of 2 μm (E). The scale bar depicted in (D) also applies for (A–C). The ciliary axoneme is stained in green by Ift88 (A) or by acetylated α-tubulin (B–E) and the BB in blue by γ-tubulin (A–E). (A–E) All measured proteins are shown in red. (A–D) *n* = 5 and (E) *n* = 3 embryos of both genotypes were used for these measurements, respectively. (A–D) At least 20 cilia per individual were used for quantification. (E) At least 10 cilia per individual were used for quantification.

F   Western blot analysis isolated from WT (*n* = 3) and *Rpgrip1l*$^{-/-}$ (*n* = 4) embryos. Actin serves as loading control.

G   qRT–PCR study to quantify *Nphp4* transcript amount in MEFs obtained from WT (*n* = 3) and *Rpgrip1l*$^{-/-}$ (*n* = 3) embryos.

Data information: The black bars represent the normalised quantification in WT MEFs and the grey bars the quantification in *Rpgrip1l*$^{-/-}$ MEFs. Data are shown as mean ± s.e.m. Statistical evaluation was performed by using unpaired Student's *t*-tests. Asterisks denote statistical significance (**P < 0.01; ***P < 0.001).

Source data are available online for this figure.

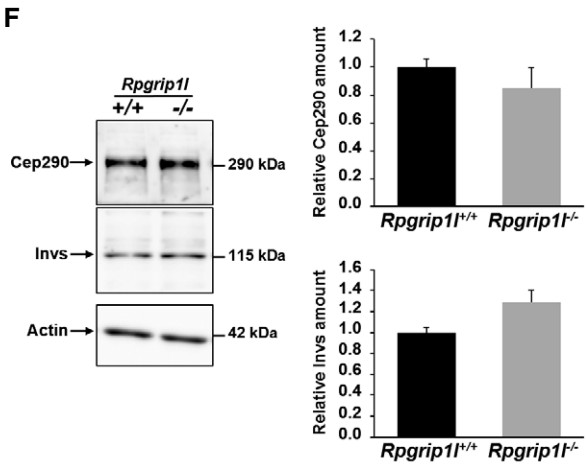

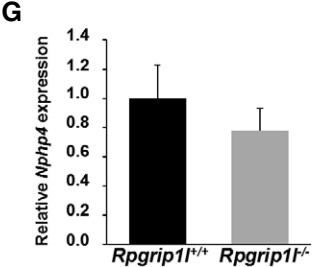

*in vivo* model for investigating functional mechanisms in regard to cilia research (Goetz & Anderson, 2010; Bimonte *et al*, 2011; Emechebe *et al*, 2016; Xin *et al*, 2017). In *Nphp4*$^{-/-}$ embryonic limb bud cilia, the TZ amount of Rpgrip1l and Cep290 was unchanged, while the amount of Nphp1 and Invs at the TZ was decreased (Fig 2F–I) confirming the data obtained in MEFs. These findings indicate that Rpgrip1l controls the TZ amount of Cep290 independent of Nphp4 in vertebrates.

To better characterise the hierarchy in TZ assembly between the proteins of the Nphp module, we inactivated Nphp1, Invs and Cep290 in NIH3T3 cells (immortalised MEFs). The CRISPR/Cas9-mediated specific inactivation of these genes was confirmed by performing several analyses in terms of on-targets and off-targets (Appendix Figs S3 and S6–S8, and Appendix Tables S1 and S2). In addition to the genotype analysis (Appendix Fig S3), we confirmed the absence of Nphp1 in *Nphp1*$^{-/-}$ NIH3T3 cells, of Invs in *Invs*$^{-/-}$ NIH3T3 cells and of Cep290 in *Cep290*$^{-/-}$ NIH3T3 cells via Western blot and immunofluorescence studies (Appendix Fig S10A–F). Previously, it was reported that the absence of Nphp1 reduces the number of ciliated cells (Delous *et al*, 2009). This was also the case in the *Nphp1*$^{-/-}$ NIH3T3 cells and re-expression of Nphp1 rescues the number of ciliated *Nphp1*$^{-/-}$ NIH3T3 cells supplying evidence of the inactivation of Nphp1 in these cells (Appendix Fig S10G–I). In *Nphp1*$^{-/-}$ NIH3T3 cells, the amount of Invs was unaltered (Fig 3A), while the amount of Nphp1 was reduced in *Invs*$^{-/-}$ NIH3T3 cells (Fig 3B). Re-expression of Invs at the ciliary TZ of *Invs*$^{-/-}$ NIH3T3 cells restored the amount of Nphp1 (Appendix Fig S10J–L) demonstrating that the TZ amount of Nphp1 depends on Invs. In *Cep290*$^{-/-}$ NIH3T3 cells, the TZ amount of Rpgrip1l was unaltered, while the amount of Nphp1, Invs and Nphp4 was decreased (Fig 3C–F) placing Cep290 between Rpgrip1l and Nphp4 in the vertebrate TZ assembly hierarchy. Ciliary length was decreased in the absence of Cep290 (Fig 3G). To evaluate the meaning of the protein axis Nphp4-Invs-Nphp1 in the development of ciliopathies caused by mutations in *Rpgrip1l*, we investigated the phenotype of *Nphp4*$^{-/-}$ mouse embryos. While the loss of Rpgrip1l leads to defects in the

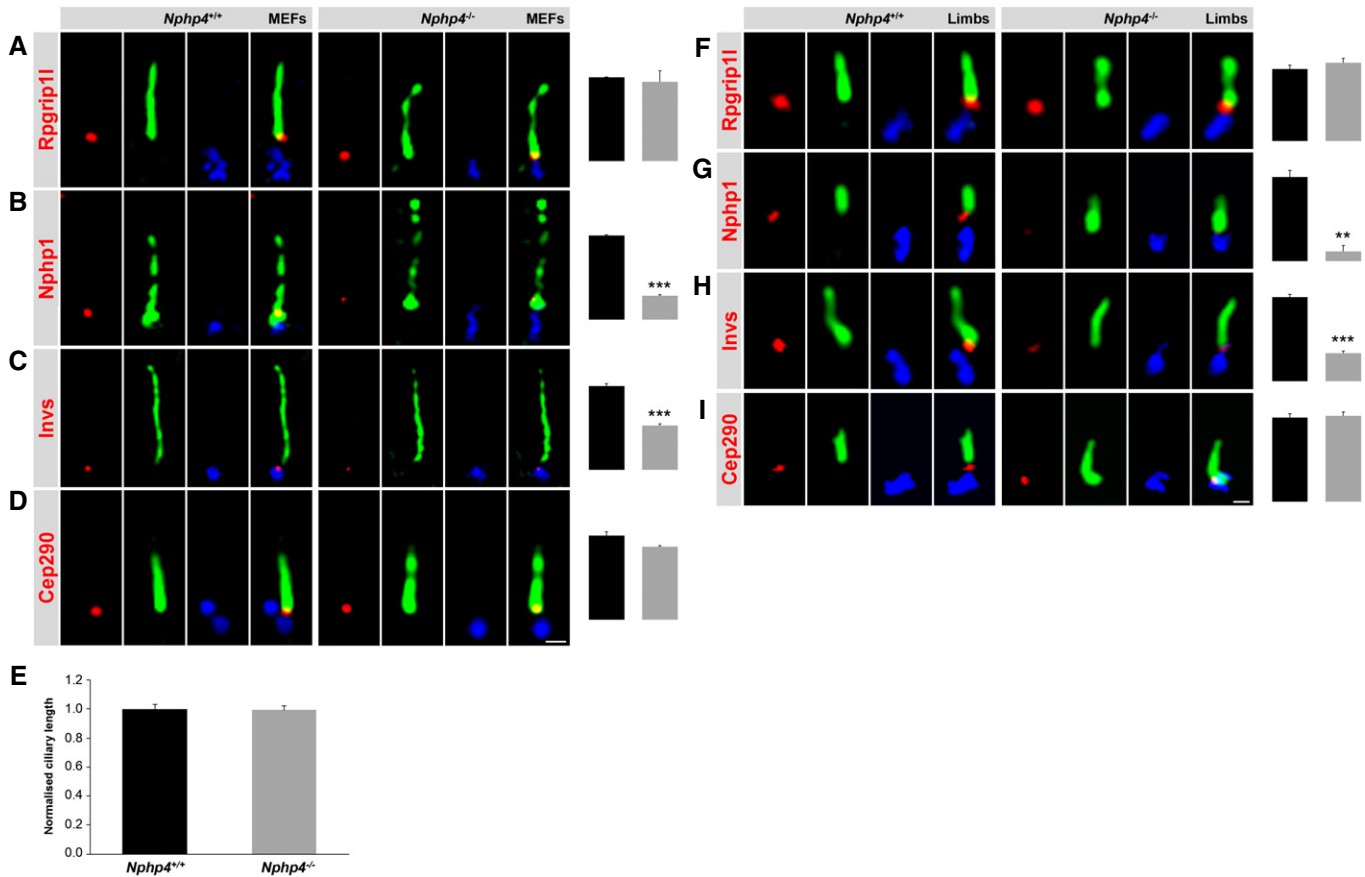

**Figure 2.   Analysis of the relationships between Nphp1, Nphp4, Invs, Cep290 and Rpgrip1l in MEFs and mouse embryonic limb buds.**

A–I   (A–D) Immunofluorescence on MEFs which were obtained from WT and *Nphp4⁻/⁻* embryos. The scale bar (in white) depicted in (D) represents a length of 0.5 μm and also applies for (A–C). (F–I) Immunofluorescence on mouse embryonic limb buds which were obtained from WT and *Nphp4⁻/⁻* embryos. The scale bar (in white) depicted in (I) represents a length of 0.5 μm and also applies for (F–H). The ciliary axoneme is stained in green by Ift88 (B), by detyrosinated α-tubulin (G) or by acetylated α-tubulin (A, C, D, F, H, I) and the BB in blue by γ-tubulin (A–D, F–I). All measured proteins are shown in red. (A and G) n = 3 and (B–E, H, I) n = 4 embryos of both genotypes were used for these measurements, respectively. (F) n = 4 WT embryos and n = 5 *Nphp4⁻/⁻* embryos were used. (E) Cilia length quantification in WT and *Nphp4⁻/⁻* MEFs. (A–I) At least 20 cilia per individual were used for quantification. (A–E) The black bar represents the normalised quantification in WT MEFs and the grey bar the quantification in *Nphp4⁻/⁻* MEFs, respectively. (F–I) The black bars represent the normalised quantification in WT mouse embryonic limb buds and the grey bars the quantification in *Nphp4⁻/⁻* mouse embryonic limb buds, respectively. (A–I) Data are shown as mean ± s.e.m. Asterisks denote statistical significance according to unpaired Student's *t*-tests (\*\*P < 0.01; \*\*\*P < 0.001).

development of, for example, the heart, the lungs, the limbs and the brain (Delous *et al*, 2007; Vierkotten *et al*, 2007; Gerhardt *et al*, 2013), we detected no alterations in these organs in *Nphp4⁻/⁻* mouse embryos compared to the WT (Fig EV2). Consequently, the very severe ciliopathy phenotype of *Rpgrip1l⁻/⁻* mouse embryos is not or not solely caused by the reduced Nphp4 amount in the absence of Rpgrip1l.

In *C. elegans*, Rpgrip1l is not only essential for the proper localisation of Nphp proteins at the TZ but also for the TZ localisation of the Mks/B9 module (Williams *et al*, 2011). Moreover, it was reported that the TZ localisation of Cep290 depends on the presence of Rpgrip1l at the TZ and that Cep290 is essential for the localisation of the Mks/B9 module members at the TZ (Li *et al*, 2016). To test whether Rpgrip1l might control the proper TZ amount of the Mks/B9 module components via ensuring the proper TZ amount of Cep290, we analysed the amount of several Mks/B9 module members at the TZ in *Rpgrip1l⁻/⁻* MEFs and

*Cep290⁻/⁻* NIH3T3 cells. The TZ amount of Mks1, Tmem67 (also known as Mks3 or Meckelin), Tctn1, Tctn2 and Tctn3 was unaffected in *Rpgrip1l⁻/⁻* MEFs (Fig EV3A–E) and in *Cep290⁻/⁻* NIH3T3 cells (Fig EV3F–J). To sum up, Rpgrip1l deficiency severely perturbs the amount of components of the Nphp modules at the TZ, but does not affect the amount of Mks/B9 module components in MEFs.

To verify the *in vitro* data obtained from *Rpgrip1l⁻/⁻* MEFs in *in vivo* systems, we tested whether Rpgrip1l is present in mouse embryonic kidneys and limb buds by performing Western blot studies. By doing so, we detected Rpgrip1l in both *in vivo* systems (Fig 4A and B). Next, we analysed the amount of components of the Nphp modules and Mks/B9 module at the TZ of cilia in *Rpgrip1l⁻/⁻* mouse embryonic kidneys and limb buds. In accordance with the *in vitro* data, we detected less Nphp1, Nphp4, Invs and Cep290 and an unaltered amount of Mks1, Tmem67, Tctn1, Tctn2 and Tctn3 at the ciliary TZ of *Rpgrip1l⁻/⁻* kidneys (Fig 4C–K). The cilia length

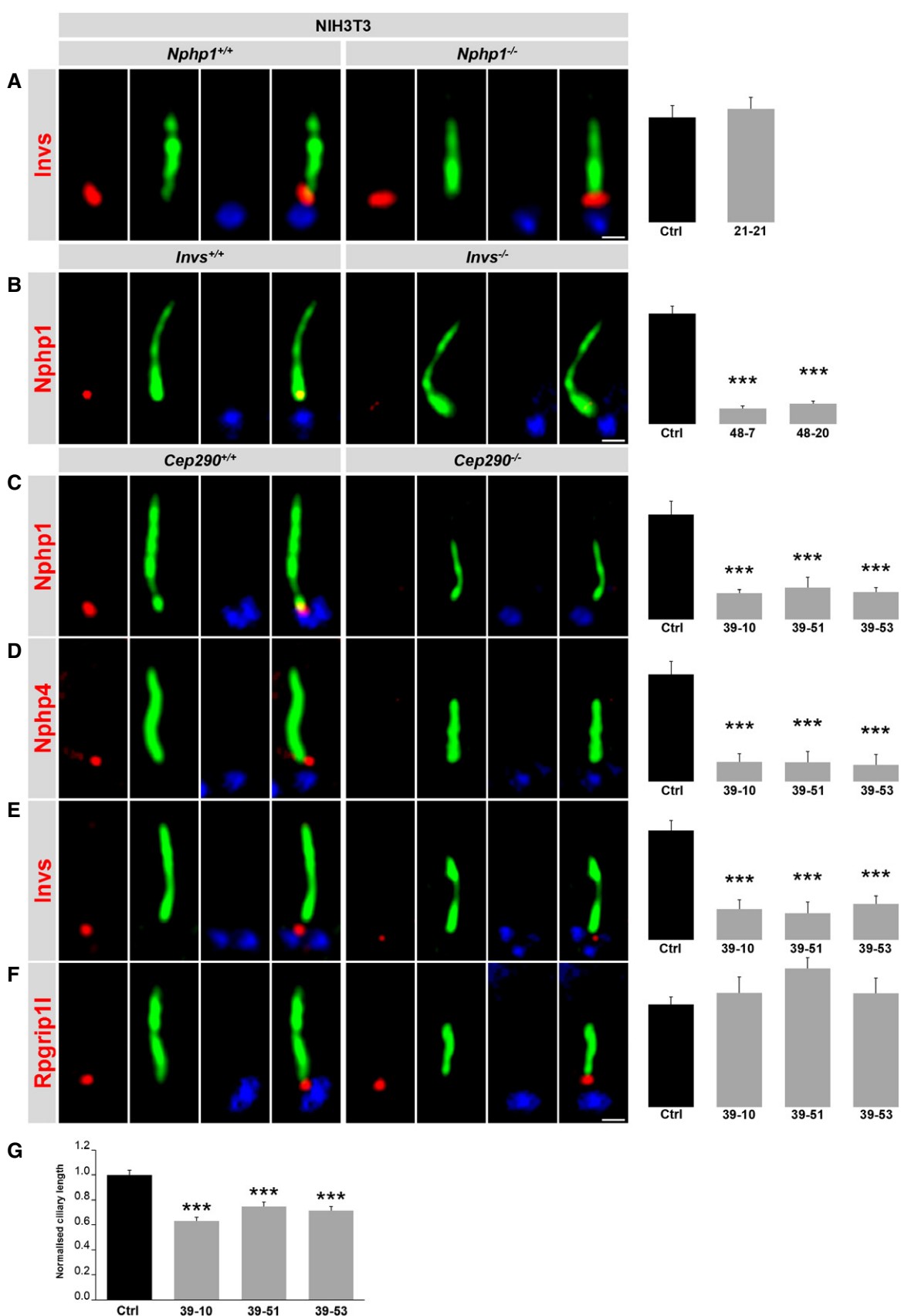

**Figure 3.**

**Figure 3. Analysis of the relationships between Nphp1, Nphp4, Invs, Cep290 and Rpgrip1l in NIH3T3 cells.**

A–G  (A–F) Immunofluorescence on WT, *Invs*$^{-/-}$, *Nphp1*$^{-/-}$ and *Cep290*$^{-/-}$ NIH3T3 cells. Different mutations of *Invs* and *Cep290* led to several clones of which each is genetically unique. The scale bars (in white) depicted in (A, B, F) represent a length of 0.5 μm and also applies for (C–E). The ciliary axoneme is stained in green by detyrosinated α-tubulin (B and C) or by acetylated α-tubulin (A, D–F) and the BB in blue by γ-tubulin (A–F). (A–F) All measured proteins are shown in red. (G) Cilia length quantification in WT and *Cep290*$^{-/-}$ NIH3T3 cells. (B–G) Different mutations of *Invs* and *Cep290* led to several clones of which each is genetically unique. (A–G) At least 20 cilia per individual were used for quantification. The black bars represent the normalised quantification in WT (Ctrl) NIH3T3 cells and the grey bars the quantification in (A) a *Nphp1*$^{-/-}$ NIH3T3 cell clone, (B) *Invs*$^{-/-}$ NIH3T3 cell clones, and (C–G) *Cep290*$^{-/-}$ NIH3T3 cell clones. The WT bars were normalised to 100%. (A–G) Data are shown as mean ± s.e.m. (A) Statistical evaluation was performed by using unpaired Student's *t*-tests. (B–G) Asterisks denote statistical significance according to one-way ANOVA and Tukey HSD tests (***$P < 0.001$; B: $F_{(2, 61)} = 17.30$, $P < 0.0001$; C: $F_{(3, 105)} = 15.18$, $P < 0.0001$; D: $F_{(3, 100)} = 16.88$, $P < 0.0001$; E: $F_{(3, 130)} = 16.75$, $P < 0.0001$; G: $F_{(3, 68)} = 21.98$, $P < 0.0001$).

was increased in *Rpgrip1l*$^{-/-}$ mouse embryonic kidneys (Fig 4L). However, the TZ amount of all these proteins was reduced in *Rpgrip1l*$^{-/-}$ embryonic limb buds (Fig 4M–U), while cilia length was increased in these limb buds (Gerhardt *et al*, 2015). Thus, the findings in embryonic limb buds differ from those obtained in MEFs and embryonic kidneys demonstrating that Rpgrip1l exerts a cell type-specific function in the murine TZ assembly hierarchy.

### Rpgrip1l controls TZ ultrastructure and TZ function as a ciliary gatekeeper in mice

In MEFs, mouse embryonic limb buds and mouse embryonic kidneys, Rpgrip1l deficiency perturbs the amount of members of the Nphp modules at the TZ (Figs 1A–E, and 4C–F and M–P), while the TZ amount of the analysed Mks/B9 proteins was unaltered in MEFs and mouse embryonic kidneys (Figs EV3A–E and 4G–K) bringing up the question whether Rpgrip1l is essential for TZ ultrastructure and TZ function as ciliary gatekeeper in these vertebrate cell types. Previously, it was shown that Rpgrip1l is arranged as discrete clusters in a ring-like fashion within the TZ (Lambacher *et al*, 2016) and a recent report proposed that the components of the Mks/B9 and Nphp modules are localised at the arms of the Y-links (Shi *et al*, 2017). We used TEM to analyse the ultrastructure of the TZ in *Rpgrip1l*$^{-/-}$ mouse embryonic kidneys and limb buds. In both, almost all Y-links were barely or not at all detectable (Fig EV4A and B). Interestingly, the electron micrographs suggest that some of the axoneme microtubules do not exist as doublets but as singlets in the absence of Rpgrip1l (Fig EV4A and B). To investigate whether TZ function is disturbed, we examined ciliary amount of the ciliary membrane protein Arl13b for which it was previously shown that its ciliary amount is reduced when the gate function of the TZ is impaired (Chih

*et al*, 2011; Garcia-Gonzalo *et al*, 2011; Roberson *et al*, 2015). The amount of Arl13b was significantly diminished within cilia of *Rpgrip1l*$^{-/-}$ MEFs, mouse embryonic kidneys and mouse embryonic limb buds (Fig EV4C–E) highlighting the importance of Rpgrip1l for TZ gatekeeper function in mice. Taken together, Rpgrip1l regulates murine TZ ultrastructure and TZ gatekeeper function independently of the Mks/B9 module.

### Synergistic function of Rpgrip1l and Rpgrip1 in the assembly of the ciliary TZ in mice

The loss of Rpgrip1l gravely perturbs vertebrate TZ protein composition and function. However, regarding its function in vertebrate TZ protein assembly, the consequence of Rpgrip1l deficiency in many vertebrate cell types is not as severe as in *C. elegans*. These data suggest the control of TZ protein composition in vertebrates by additional, vertebrate-specific factors. In basal deuterostomes and other invertebrate species like *C. elegans*, a single orthologue of Rpgrip1l is present, while in vertebrates an additional paralogue, Rpgrip1, exists (Vierkotten *et al*, 2007; Fig 5A). Rpgrip1 was present at the basal body (BB) of MEF cilia and localised in close proximity (nearer than 40 nm) to Rpgrip1l (Fig 5B–D, and Appendix Fig S11A and B). Intriguingly, its amount was significantly increased at the BB of *Rpgrip1l*$^{-/-}$ MEFs (Fig 5E), although the overall cellular amount of Rpgrip1 was unaltered in *Rpgrip1l*$^{-/-}$ MEFs (Fig 5F) demonstrating that Rpgrip1l does not control the Rpgrip1 amount *per se* but the amount of Rpgrip1 at the BB. To test whether Rpgrip1 is also present in our *in vivo* systems, we performed Western blot studies with protein lysates obtained from WT mouse embryonic kidneys and limb buds. In this way, we detected Rpgrip1 in both *in vivo* systems (Fig 5G and H). The amount of Rpgrip1 was increased at the BB of *Rpgrip1l*$^{-/-}$ mouse embryonic kidneys but

**Figure 4. Rpgrip1l deficiency affects the assembly of the vertebrate TZ in a cell type-specific manner.**

A, B  Western blot analysis for Rpgrip1l. Lysates isolated from (A) WT embryonic kidneys at embryonic stage E18.5 ($n = 6$) and (B) WT embryonic limb buds at embryonic stage E12.5 ($n = 3$). Actin serves as loading control.

C–U  (C–K, M–U) Immunofluorescence on mouse embryonic kidneys (C–K) and mouse embryonic limb buds (M–U) which were obtained from WT and *Rpgrip1l*$^{-/-}$ embryos. The scale bar (in white) depicted in (K) represents a length of 0.5 μm and also applies for (C–J). The scale bar (in white) depicted in (U) represents a length of 0.5 μm and also applies for (M–T). The ciliary axoneme is stained in green by detyrosinated α-tubulin (C and M) or by acetylated α-tubulin (D–K, N–U) and the BB in blue by γ-tubulin (C–K, M–U). (C–K, M–U) All measured proteins are shown in red. (L) Cilia length quantification in WT and *Rpgrip1l*$^{-/-}$ mouse embryonic kidneys. (C–L) $n = 3$ and (M–Q) $n = 4$ embryos of both genotypes were used for these measurements, respectively. (R, S, U) $n = 4$ WT embryos and $n = 5$ *Rpgrip1l*$^{-/-}$ embryos were used. (T) $n = 3$ WT embryos and $n = 5$ *Rpgrip1l*$^{-/-}$ embryos were used. (C–U) At least 20 cilia per individual were used for quantification. The black bars represent the normalised quantification in WT embryonic kidneys and limb buds and the grey bars the quantification in *Rpgrip1l*$^{-/-}$ embryonic kidneys and limb buds. Data are shown as mean ± s.e.m. Asterisks denote statistical significance according to unpaired Student's *t*-tests (*$P < 0.05$; **$P < 0.01$; ***$P < 0.001$).

Source data are available online for this figure.

not in *Rpgrip1l⁻/⁻* limb buds (Fig 5I and J) indicating that Rpgrip1l deficiency affects the Rpgrip1 amount at the BB in a cell type-specific manner. Interestingly, the TZ amount of Rpgrip1l was unaltered in *Rpgrip1^{nmf247/nmf247}* MEFs (Fig 5K).

Assuming that the Rpgrip1 amount at the BB of *Rpgrip1l⁻/⁻* MEF and mouse embryonic kidney cilia is elevated in order to

compensate for the loss of Rpgrip1l, we examined the ciliary localisation of TZ proteins and the ciliary membrane-associated protein Arl13b in *Rpgrip1^{nmf247}* mutant and *Rpgrip1l; Rpgrip1^{nmf247}*-combined mutant MEFs. In *Rpgrip1^{nmf247/nmf247}* MEFs, the amount of Nphp1, Nphp4, Invs and Cep290 was reduced (Fig 6A–D) indicating that Rpgrip1 participates in organising the composition of the

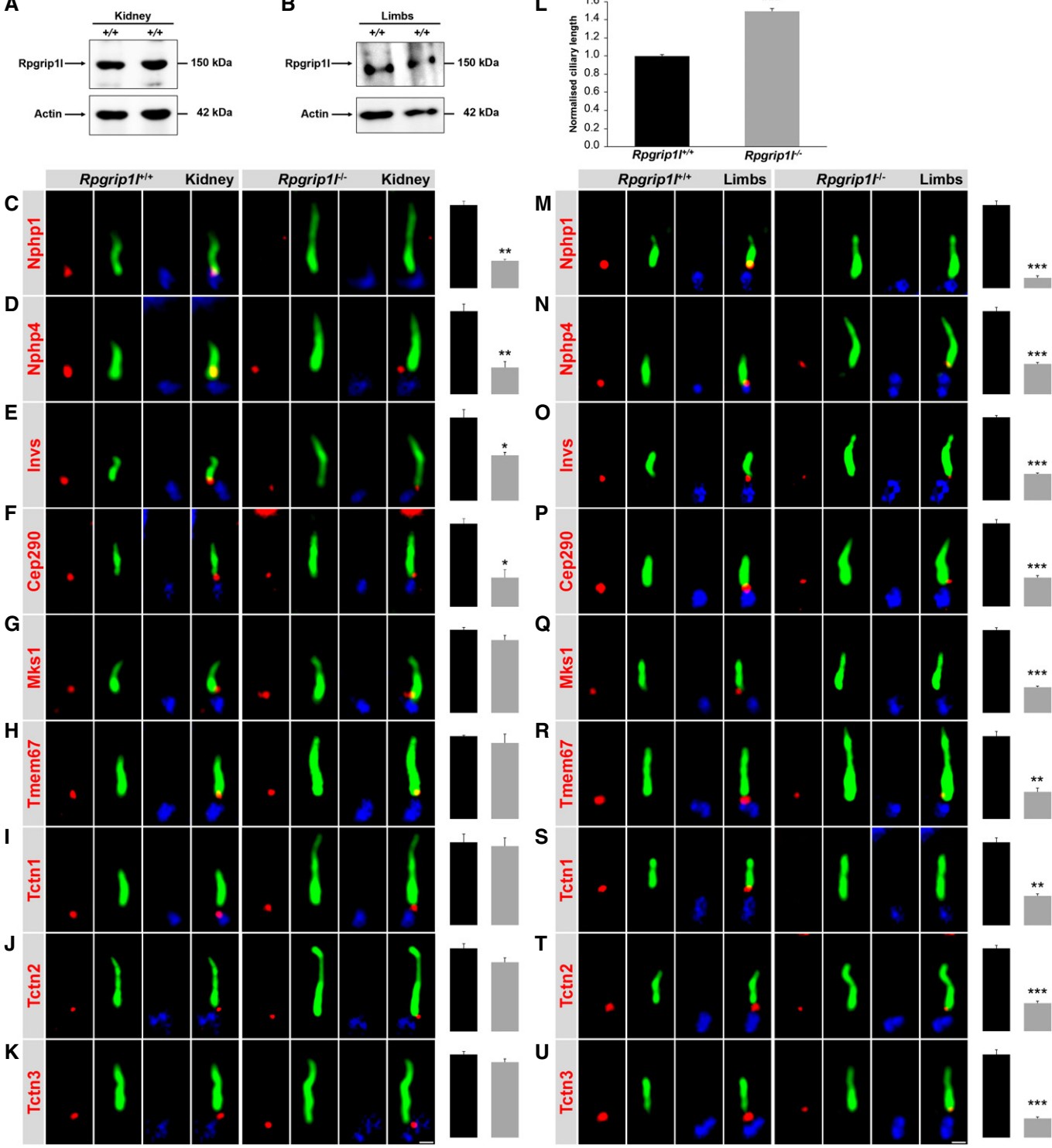

**Figure 4.**

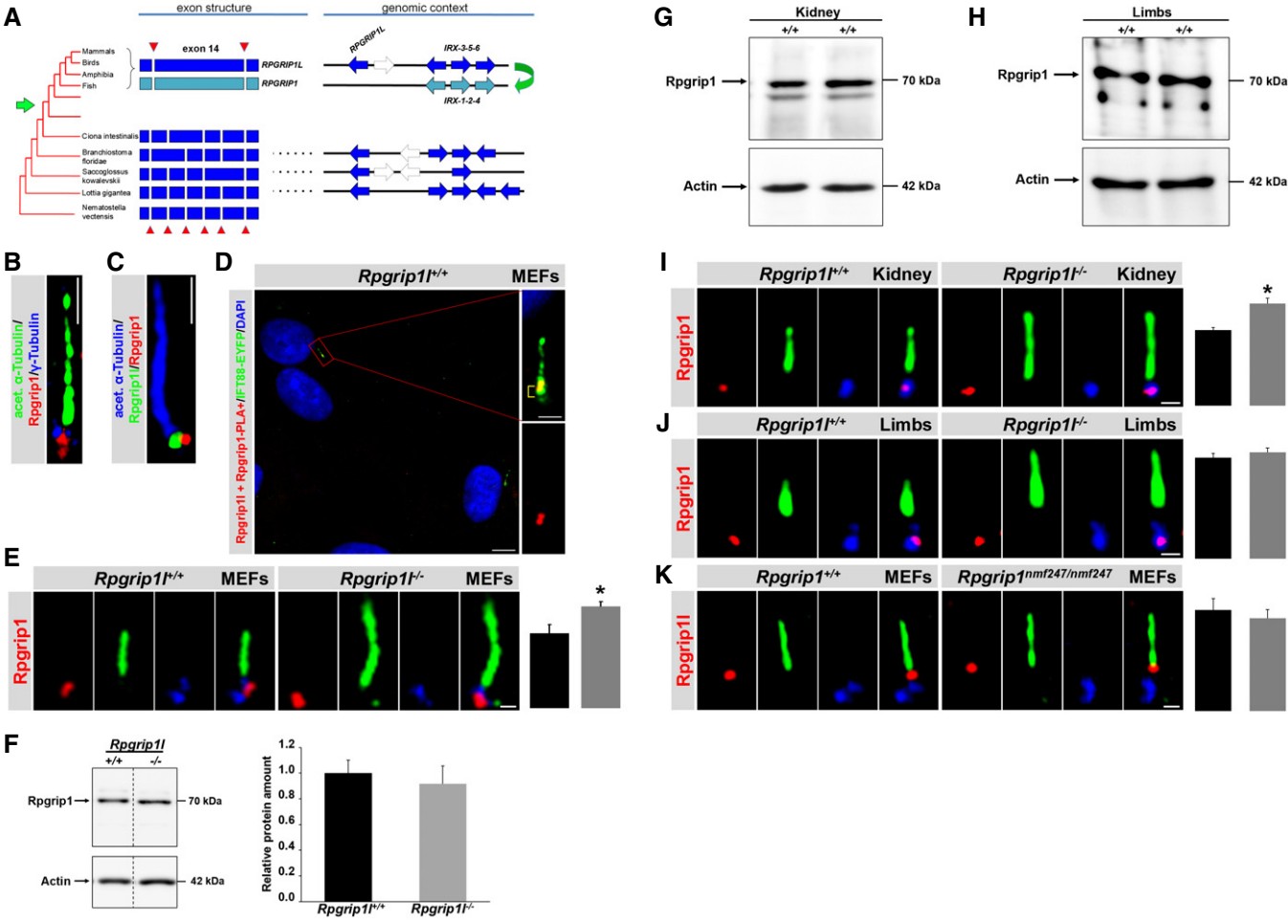

**Figure 5. The amount of the Rpgrip1l-related protein Rpgrip1 is affected by Rpgrip1l deficiency in a cell type-specific manner.**

A Inspection of exon boundaries of human *RPGRIP1L* and *RPGRIP1* genes reveals that exon 14 is an apparent fusion product of several smaller exons existing in basal deuterostomes and other invertebrate species. Thus, *RPGRIP1L* and *RPGRIP1* arose by duplication at the base of the vertebrate lineage (green arrow at species tree). Vertebrate *RPGRIP1L* (shown for human) is located near a cluster of three homeobox genes (*IRX3-5-6*). Synteny analysis in *Branchiostoma floridae* (NW_003101557), *Saccoglossus kowalevskii* (NW_003137831) and *Lottia gigantea* (Kerner *et al*, 2009) shows that this is an ancient arrangement. Other (orthologous) genes in genomic intervals, not considered here, are shown in white arrows. As a result of whole genome duplications early in the history of vertebrates (green arrows), there exists a second paralogous *IRX* cluster (*IRX1-2-4*) in vertebrates, from which probably also *RPGRIP1* originated. Subsequently, *RPGRIP1* then became transposed to a different chromosomal location (not shown here).

B–K (B–E, I–K) Immunofluorescence on MEFs (B–E, K), mouse embryonic kidneys (I) and mouse embryonic limb buds (J) obtained from WT (B–E and I–K), *Rpgrip1l*$^{-/-}$ (E, I, J) and *Rpgrip1*$^{nmf247/nmf247}$ embryos (K). (B and C) Micrographs were acquired using super-resolution microscopy (3D-SIM). (B, C, E, I–K) The scale bars (in white) represent a length of 0.5 μm. (D) Scale bars (in white) depict 10 μm (overview) and 1 μm (magnification). The ciliary axoneme is stained in green (B, E, I–K) or in blue (C) by acetylated α-tubulin, Rpgrip1 in red (B, C, E, I, J), Rpgrip1l in green (C) or in red (K), and the BB in blue by γ-tubulin (B, E, I–K). (K) Rpgrip1l is shown in red. (E) *n* = 4, (I) *n* = 3 and (K) *n* = 5 embryos of both genotypes were used. (J) *n* = 4 WT embryos and *n* = 5 *Rpgrip1l*$^{-/-}$ embryos were used. (E, I–K) At least 20 cilia per individual were used for quantification. The black bars represent the quantification in the WT, the grey bars the quantification in the mutant. The WT bars were normalised to 100%. Data are shown as mean ± s.e.m. Asterisks denote statistical significance according to unpaired Student's *t*-tests (**P* < 0.05). (D) *In situ* proximity ligation assay (*in situ* PLA) on MEFs. Cell nuclei are marked by DAPI (in blue), the ciliary axoneme by transiently transfected Ift88-EYFP (in green). The PLA signal is shown in red. In the magnification, additional accumulation of Ift88-EYFP at the ciliary base is highlighted by yellow brackets. (F) Western blot analysis with lysates isolated from WT (*n* = 3) and *Rpgrip1l*$^{-/-}$ (*n* = 4) embryos. (G, H) Western blot analysis with lysates obtained from WT mouse embryonic kidneys (*n* = 6) and from WT limb buds (*n* = 3). (F–H) Actin serves as loading control.

Source data are available online for this figure.

mouse TZ. However, this reduction was not as strong as in *Rpgrip1l*$^{-/-}$ MEFs (Fig 1A–D). The amount of all analysed Mks/B9 module components at the TZ was unaltered in *Rpgrip1*$^{nmf247/nmf247}$ MEFs (Fig 6E–I), like in *Rpgrip1l*$^{-/-}$ MEFs (Fig EV3A–E). The ciliary amount of Arl13b was also unchanged in *Rpgrip1*$^{nmf247/nmf247}$ MEFs (Fig 6J) reflecting that Rpgrip1 deficiency does not affect the function

of the TZ as a ciliary gate. Moreover, the cilia length was unaffected in *Rpgrip1*$^{nmf247/nmf247}$ MEFs and elevated in *Rpgrip1l*$^{-/-}$; *Rpgrip1*$^{nmf247/nmf247}$ MEFs (Fig 6K). In *Rpgrip1l*$^{-/-}$; *Rpgrip1*$^{nmf247/nmf247}$ MEFs, the amount of the analysed components of the Nphp modules at the TZ was decreased in a similar manner as in *Rpgrip1l*$^{-/-}$ MEFs (Figs 1A–E and 6A–D). Importantly, the amount of all analysed

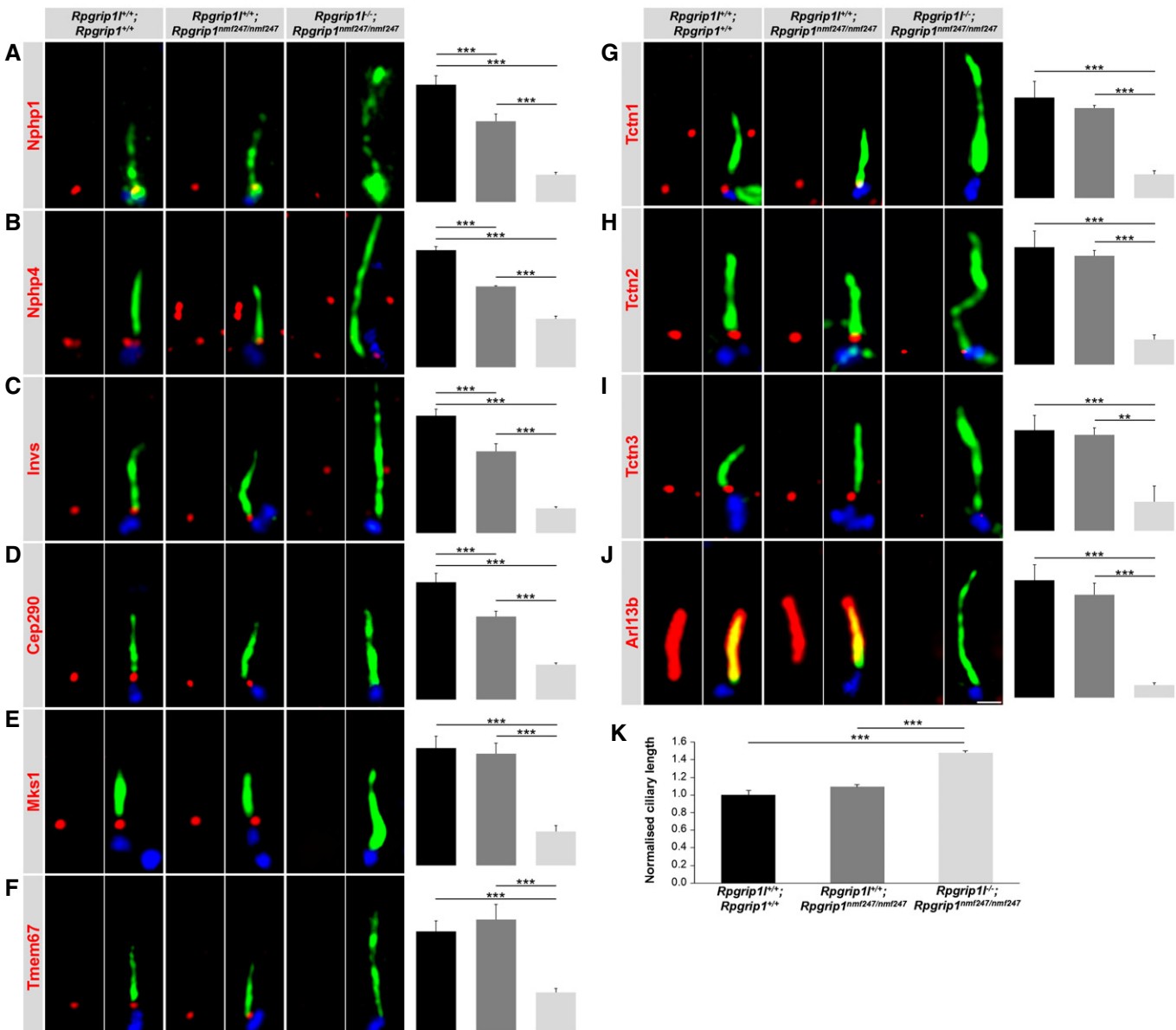

**Figure 6. Rpgrip1l and Rpgrip1 synergistically regulate the TZ amounts of Mks/B9 module components in MEFs.**

A–K  Immunofluorescence on MEFs obtained from WT, *Rpgrip1^{nmf247/nmf247}* and *Rpgrip1l^{−/−}*; *Rpgrip1^{nmf247/nmf247}* embryos. Scale bar (in white) represents a length of 0.5 μm. The scale bar depicted in (J) also applies for (A–I). The ciliary axoneme is stained in green by Ift88 (A) or by acetylated α-tubulin (B–J) and the BB in blue by γ-tubulin (A–J). (A–J) All measured proteins are shown in red. (K) Cilia length quantification in WT, *Rpgrip1^{nmf247/nmf247}* and *Rpgrip1l^{−/−}*; *Rpgrip1^{nmf247/nmf247}* MEFs. (A–K) At least 20 cilia per individual were used for quantification. The embryo sample size is as follows: (A) WT: 5, *Rpgrip1^{nmf247/nmf247}*: 6, *Rpgrip1l^{−/−}*; *Rpgrip1^{nmf247/nmf247}*: 6; (B) WT: 4, *Rpgrip1^{nmf247/nmf247}*: 4, *Rpgrip1l^{−/−}*; *Rpgrip1^{nmf247/nmf247}*: 4; (C, H) WT: 6, *Rpgrip1^{nmf247/nmf247}*: 6, *Rpgrip1l^{−/−}*; *Rpgrip1^{nmf247/nmf247}*: 6; (D) WT: 5, *Rpgrip1^{nmf247/nmf247}*: 5, *Rpgrip1l^{−/−}*; *Rpgrip1^{nmf247/nmf247}*: 6; (E, F, K) WT: 4, *Rpgrip1^{nmf247/nmf247}*: 5, *Rpgrip1l^{−/−}*; *Rpgrip1^{nmf247/nmf247}*: 4; (G) WT: 3, *Rpgrip1^{nmf247/nmf247}*: 5, *Rpgrip1l^{−/−}*; *Rpgrip1^{nmf247/nmf247}*: 4; (I) WT: 5, *Rpgrip1^{nmf247/nmf247}*: 3, *Rpgrip1l^{−/−}*; *Rpgrip1^{nmf247/nmf247}*: 5; (J) WT: 6, *Rpgrip1^{nmf247/nmf247}*: 5, *Rpgrip1l^{−/−}*; *Rpgrip1^{nmf247/nmf247}*: 3. (A–K) The black bars represent the quantification in WT MEFs, the dark grey bars the quantification in *Rpgrip1^{nmf247/nmf247}* MEFs and the bright grey bars the quantification in *Rpgrip1l^{−/−}*; *Rpgrip1^{nmf247/nmf247}* MEFs. The WT bars were normalised to 100%. Data are shown as mean ± s.e.m. Asterisks denote statistical significance according to one-way ANOVA and Tukey HSD tests (*$P < 0.05$; **$P < 0.01$; ***$P < 0.001$) (A: $F_{(2, 57)}$ = 96.49, $P < 0.0001$; B: $F_{(2, 57)}$ = 261.1, $P < 0.0001$; C: $F_{(2, 57)}$ = 100.1, $P < 0.0001$; D: $F_{(2, 57)}$ = 192.0, $P < 0.0001$; E: $F_{(2, 106)}$ = 25.33, $P < 0.0001$; F: $F_{(2, 107)}$ = 27.47, $P < 0.0001$; G: $F_{(2, 27)}$ = 21.68, $P < 0.0001$; H: $F_{(2, 111)}$ = 22.93, $P < 0.0001$; I: $F_{(2, 92)}$ = 9.682, $P < 0.0001$; J: $F_{(2, 129)}$ = 98.1, $P < 0.0001$; K: $F_{(2, 151)}$ = 78.91, $P < 0.0001$).

Mks/B9 module components at the TZ was reduced in *Rpgrip1l^{−/−}*; *Rpgrip1^{nmf247/nmf247}* MEFs (Fig 6E–I) demonstrating that Rpgrip1l and Rpgrip1 synergistically control the amount of the analysed Mks/B9 module components at the murine TZ.

Our analyses of the TZ protein composition in MEFs revealed a synergistical function of Rpgrip1l and Rpgrip1 in regulating TZ assembly. To analyse whether this synergistical function is reflected in the phenotype of *Rpgrip1l^{−/−}*; *Rpgrip1^{nmf247/nmf247}* combined

mutant mouse embryos, we turned our attention to morphological and histological examinations. In comparison to $Rpgrip1l^{-/-}$ mouse embryos, the combined mutants display a higher frequency of exencephaly, eye malformations and ventricular septal defects (Fig 7A) indicating that Rpgrip1l and Rpgrip1 cooperate to ensure proper murine brain, eye and heart development. In the case of the ventricular septal defects, the difference is even statistically significant. Furthermore, while $Rpgrip1^{nmf247/nmf247}$ and $Rpgrip1l^{+/-}$ mouse embryos do not show any defects (Vierkotten et al, 2007; Won et al, 2009; Gerhardt et al, 2013), their combined mutations induce polycystic kidneys in $Rpgrip1l^{+/-}$; $Rpgrip1^{nmf247/nmf247}$ embryos (Fig 7B and C) demonstrating a functional synergy of Rpgrip1l and Rpgrip1 in renal development. In this context, we investigated the TZ ultrastructure in $Rpgrip1l^{-/-}$; $Rpgrip1^{nmf247/nmf247}$ embryonic kidneys. While we observed a dramatically reduced number of Y-links in $Rpgrip1l^{-/-}$ kidneys, we were not able to identify any TZ in the four investigated $Rpgrip1l^{-/-}$; $Rpgrip1^{nmf247/nmf247}$ kidneys. Neither we found Y-links nor any clearly detectable ciliary pocket region indicating that the ultrastructure of the TZ might be much more affected than the ultrastructure in $Rpgrip1l^{-/-}$ kidneys.

**Synergistic function of Rpgrip1l and Rpgrip1 in the assembly of the ciliary TZ in HEK293 cells**

Moreover, we investigated the role of Rpgrip1l as well as of Rpgrip1l and Rpgrip1 in TZ assembly in the human cell line HEK293 (human embryonic kidney 293). For this purpose, we generated several $RPGRIP1L^{-/-}$ and $RPGRIP1L^{-/-}$; $RPGRIP1^{-/-}$ HEK293 cell clones by using the CRISPR/Cas9 system (Appendix Figs S3–S5, S9 and S12A–D, and Appendix Tables S1 and S2). In $RPGRIP1L^{-/-}$ HEK293 cell clones, we analysed the amount of some TZ proteins exemplarily. The amount of Nphp1, Invs and Cep290 was decreased at the ciliary TZ (Fig 8A–C), while the TZ amount of Tctn2 was unaltered (Fig 8D). Exogenous re-expression of Rpgrip1l via transfection of pRpgrip1l into $RPGRIP1L^{-/-}$ HEK293 cells restored the TZ amount of Nphp1 demonstrating that Rpgrip1l controls the amount of Nphp1 at the ciliary TZ in HEK293 (Appendix Fig S12E–G). Thus, the results obtained from $RPGRIP1L^{-/-}$ HEK293 cells resemble those provided by $Rpgrip1l^{-/-}$ mouse embryonic kidneys and MEFs (Figs 1A, C and D, 4C, E, F and J, 8A–D, and EV3D). Moreover, the cilia length of $RPGRIP1L^{-/-}$ HEK293 cells was increased (Fig 8E) according to the data obtained from $Rpgrip1l^{-/-}$ murine embryonic kidneys (Fig 4L). To evaluate whether Rpgrip1 plays the same role in the human TZ composition as in the murine TZ composition, we investigated double mutant HEK293 cell clones. In $RPGRIP1L^{-/-}$; $RPGRIP1^{-/-}$ HEK293 cells, the TZ amount of all analysed TZ proteins was reduced (Fig 8A–D). Exogenous re-expression of Rpgrip1l via transfection of pRpgrip1l into $RPGRIP1L^{-/-}$; $RPGRIP1^{-/-}$ HEK293 cells increased but did not restore the TZ amount of Nphp1 (Appendix Fig S12H–K) reflecting the loss of Rpgrip1. Furthermore, the cilia length of double mutant HEK293 cells was elevated in comparison to WT HEK293 cells but unchanged in comparison to $RPGRIP1L^{-/-}$ HEK293 cells (Fig 8E).

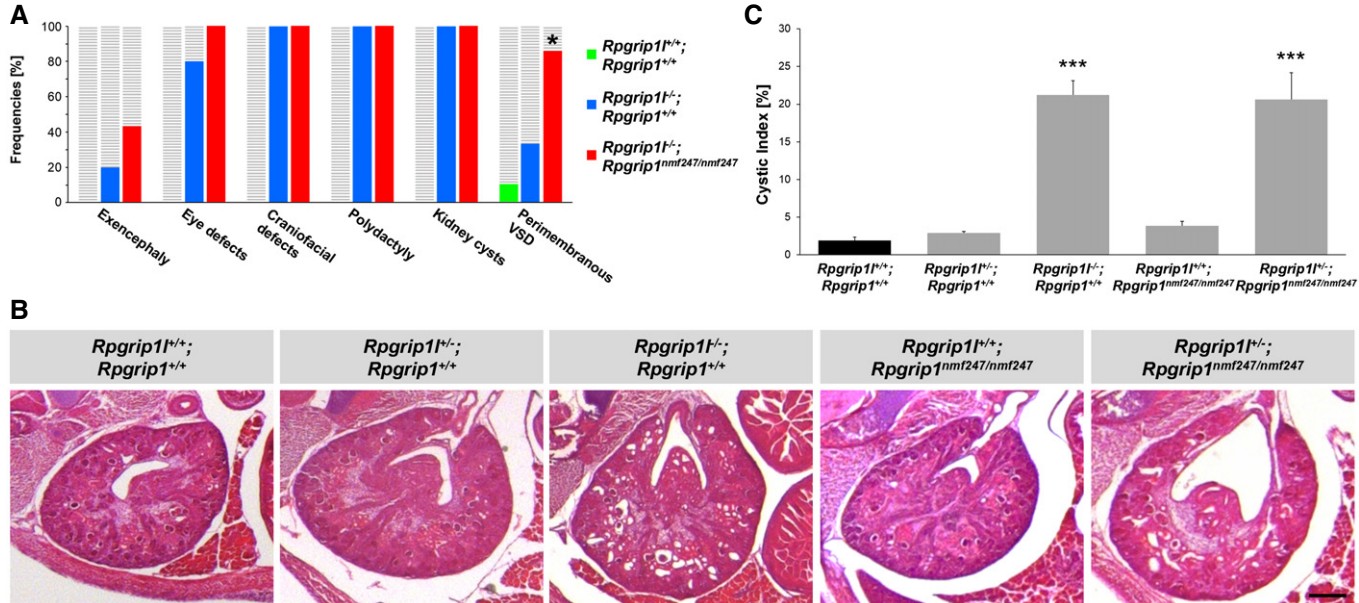

**Figure 7. Rpgrip1l and Rpgrip1 synergistically regulate the development of several organs in mouse embryos.**

A    Illustration of several defect frequencies in WT, $Rpgrip1l^{-/-}$ and $Rpgrip1l^{-/-}$; $Rpgrip1^{nmf247/nmf247}$ mouse embryos. $n = 10$ WT, $n = 15$ (in case of the kidney: $n = 9$) $Rpgrip1l^{-/-}$ and $n = 7$ $Rpgrip1l^{-/-}$; $Rpgrip1^{nmf247/nmf247}$ embryos were used. Asterisk denotes statistical significance between the defect frequency in $Rpgrip1l^{-/-}$; $Rpgrip1^{nmf247/nmf247}$ and $Rpgrip1l^{-/-}$ hearts according to chi-square test ($*P < 0.05$).

B, C    Haematoxylin and eosin (H & E) staining (B) and cystic index analysis (C) of mouse embryonic kidneys (at E18.5) in WT, $Rpgrip1l^{+/-}$, $Rpgrip1l^{-/-}$, $Rpgrip1^{nmf247/nmf247}$ and $Rpgrip1l^{-/-}$; $Rpgrip1^{nmf247/nmf247}$ mouse embryos. $n = 10$ WT, $n = 21$ $Rpgrip1l^{+/-}$, $n = 9$ $Rpgrip1l^{-/-}$, $n = 8$ $Rpgrip1^{nmf247/nmf247}$ and $n = 7$ $Rpgrip1l^{+/-}$; $Rpgrip1^{nmf247/nmf247}$ mouse embryonic kidneys were investigated. The scale bar (in black) represents a length of 250 μm and applies for all pictures. (C) Cystic indices were calculated on representative renal sections as a measure of severity of polycystic kidney disease. The black bar represents the normalised quantification in WT kidneys and the grey bars the quantification in the mutant kidneys. Data are shown as mean ± s.e.m. Asterisks denote statistical significance according to one-way ANOVA and Tukey HSD tests ($***P < 0.001$; $F(4, 18) = 66.64$, $P < 0.00001$).

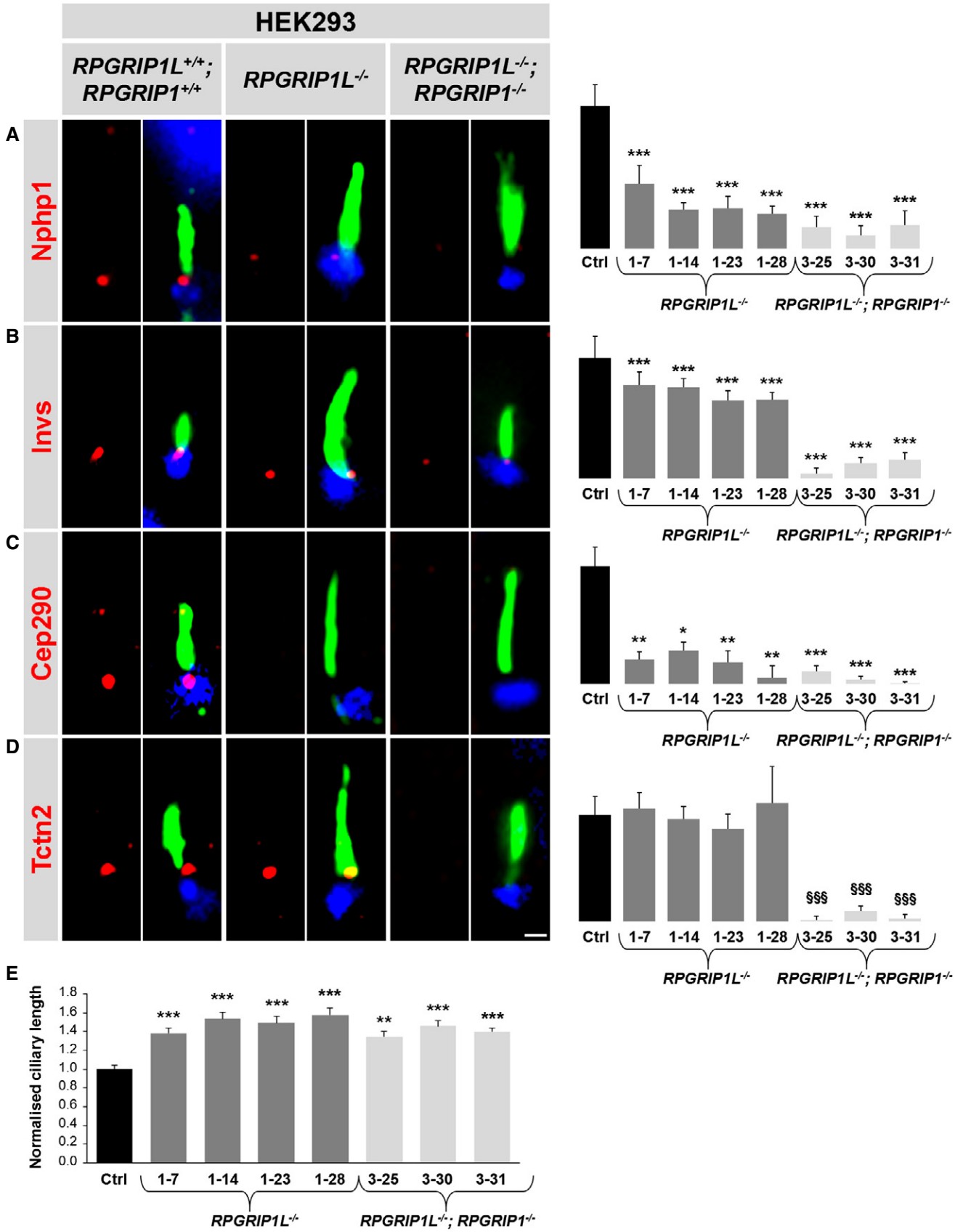

Figure 8.

**Figure 8.  RPGRIP1L and RPGRIP1 are essential for TZ assembling in human embryonic kidney cells.**

A–D    Immunofluorescence on WT, *RPGRIP1L*$^{-/-}$ and *RPGRIP1L*$^{-/-}$; *RPGRIP1*$^{-/-}$ HEK293 cells. Different mutations of *RPGRIP1L* and *RPGRIP1* led to several clones of which each is genetically unique. The scale bar (in white) depicted in (D) represents a length of 0.5 μm and also applies for (A–C). The ciliary axoneme is stained in green by detyrosinated α-tubulin (A) or by acetylated α-tubulin (B–D) and the BB in blue by γ-tubulin (A–D). (A–D) All measured proteins are shown in red.

E     Cilia length quantification in WT, *RPGRIP1L*$^{-/-}$ and *RPGRIP1L*$^{-/-}$; *RPGRIP1*$^{-/-}$ HEK293 cells.

Data information: Different mutations of *RPGRIP1L* and *RPGRIP1* led to several clones of which each is genetically unique. The black bars represent the normalised quantification in WT (Ctrl) HEK293 cells, the dark grey bars the quantification in several *RPGRIP1L*$^{-/-}$ HEK293 cell clones and the bright grey bars the quantification in several *RPGRIP1L*$^{-/-}$; *RPGRIP1*$^{-/-}$ HEK293 cell clones. At least 10 cilia per individual were used for quantification. Data are shown as mean ± s.e.m. Asterisks and section signs denote statistical significance according to one-way ANOVA and Tukey HSD tests (*$P < 0.05$; **$P < 0.01$; ***$P < 0.001$; §§§$P < 0.001$) [A: $F_{(7, 112)} = 14.90$, $P < 0.0001$; B: $F_{(7, 118)} = 18.30$, $P < 0.0001$; C: $F_{(7, 118)} = 15.7$, $P < 0.0005$; D: $F_{(7, 112)} = 16.37$, $P < 0.0001$; E: $F_{(7, 312)} = 8.836$, $P < 0.0001$]. Asterisks indicate significance in comparison to the WT, while section signs represent significance in comparison to the WT and the *RPGRIP1L*$^{-/-}$ clones.

## Discussion

Ciliopathies are life-threatening diseases and the number of severe diseases shown or suggested to be ciliopathies is continually increasing (Reiter & Leroux, 2017). Since many human ciliopathies are caused by mutations in genes encoding TZ proteins (Hildebrandt *et al*, 2011; Czarnecki & Shah, 2012), the understanding of TZ assembly and organisation is of an enormous importance. Many investigations which dealt with this topic were performed in *C. elegans* where a stringent TZ assembly hierarchy was described with Rpgrip1l at its top (Williams *et al*, 2011; Jensen *et al*, 2015; Li *et al*, 2016). However, little is known about the functional mechanisms of TZ assembling in vertebrates. In our study, we focussed on the role of Rpgrip1l in vertebrate TZ assembling. Previously, it was demonstrated that Rpgrip1l is essential for the proper TZ localisation of Nphp1, Nphp4 and Cep290 in *C. elegans* (Williams *et al*, 2011; Jensen *et al*, 2015; Li *et al*, 2016). Our investigations reveal that Rpgrip1l regulates the amount of these members of the Nphp modules at the TZ of murine cilia (Figs 1A–E, and 4C–F and M–P).

By performing quantifications of Nphp protein amounts at the ciliary TZ of *Nphp4*$^{-/-}$ MEFs and mouse embryonic limbs, we found out that Nphp4 which directly interacts with Rpgrip1l and Nphp1 (Mollet *et al*, 2002) occupies a position between Rpgrip1l and the two proteins Nphp1 and Invs in the vertebrate TZ assembly hierarchy (Figs 2A–C and F–I, and 9A). Thus, Rpgrip1l seems to control the proper amount of Nphp1 and Invs at the vertebrate TZ by controlling the TZ amount of Nphp4. Moreover, it was reported before that Nphp1 directly interacts with Invs (Otto *et al*, 2003) and our data show that Invs outranks Nphp1 in the vertebrate TZ assembly hierarchy (Figs 3A and B, and 9A). Strikingly, Nphp1 localises independently of Invs to the TZ in *C. elegans* (Warburton-Pitt *et al*, 2012). In addition, Rpgrip1l deficiency does not decrease the amount of Invs at the ciliary TZ in *C. elegans* but expands the length of the Invs signal (Jensen *et al*, 2015). Consequently, these findings represent obvious differences between the TZ organisation of *C. elegans* and mice. Since *Nphp1-* and *Nphp4*-mutant mice display only eye and male reproductive organ defects (Jiang *et al*, 2008, 2009; Louie *et al*, 2010; Won *et al*, 2011) and *Invs*-mutant mouse

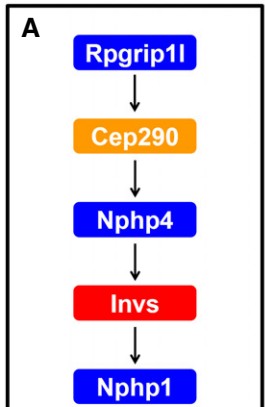
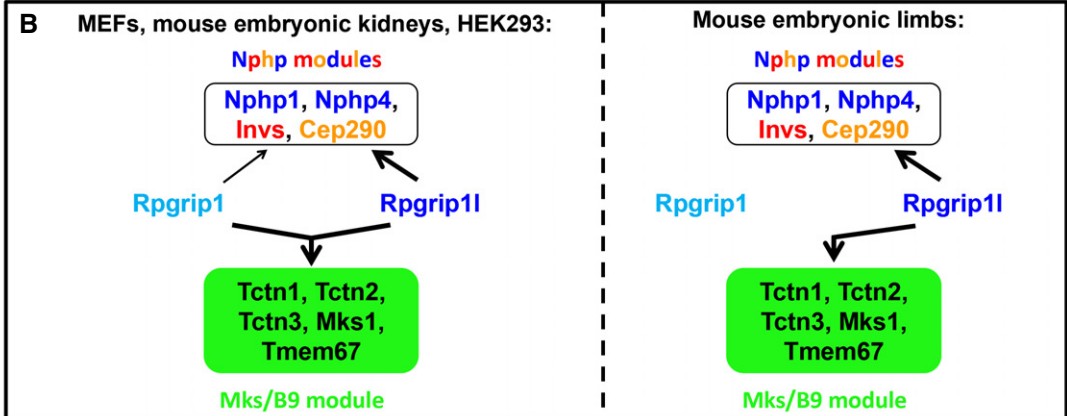

**Figure 9.   Schematic illustration of the TZ assembly organisation in vertebrates.**

A    Proposed model of the organisation of the Rpgrip1l-Cep290-Nphp4-Invs-Nphp1 axis. The arrows indicate the quantitative dependency between the TZ proteins. Rpgrip1l regulates the amount of Cep290 at the vertebrate TZ. In turn, Cep290 controls the TZ amount of Nphp4 which ensures the proper amount of Invs at the TZ. Finally, Invs governs the TZ amount of Nphp1.

B    Two schemes that depict the cell type-specific roles of Rpgrip1l and Rpgrip1 in the vertebrate TZ assembly hierarchy. In MEFs, mouse embryonic kidneys and HEK293 cells, each of both proteins, Rpgrip1l and Rpgrip1, controls the TZ amount of all analysed components of the Nphp modules. However, the impact of Rpgrip1l on the Nphp proteins is much stronger than that of Rpgrip1 (indicated by the different size of the arrows). In contrast to the Nphp modules, Rpgrip1l and Rpgrip1 synergistically regulate the TZ amount of the Mks/B9 module members (represented by the fused arrow). In mouse embryonic limb buds, Rpgrip1l alone is able to govern the TZ amount of both, the components of the Nphp modules and the components of the Mks/B9 module.

Data information: Colour code: blue, Nphp1-4-8 module components; red, Invs compartment components; orange, Nphp5-6 module components; green, Mks/B9 module components.

embryos exhibit kidney cysts and left–right asymmetry defects but, for example, no brain defects (Mochizuki *et al*, 1998; Morgan *et al*, 1998), we deduce that the very severe ciliopathy phenotype of $Rpgrip1l^{-/-}$ mouse embryos is not or not completely caused by the impact of Rpgrip1l deficiency on the Nphp4-Invs-Nphp1 axis. Thus, at least the brain defects of $Rpgrip1l^{-/-}$ mouse embryos which resemble malformations found in human Meckel and Joubert syndrome patients arise independent of the impaired Nphp4-Invs-Nphp1 axis.

Interestingly, *Cep290*-mutant mouse embryos display many features which resemble symptoms of Meckel and Joubert syndrome patients. These embryos suffer from brain, eye, kidney and heart defects (Chang *et al*, 2006; McEwen *et al*, 2007; Lancaster *et al*, 2011; Li *et al*, 2015; Rachel *et al*, 2015) and hence show a mutant phenotype similar to that of $Rpgrip1l^{-/-}$ mouse embryos (Delous *et al*, 2007; Vierkotten *et al*, 2007; Besse *et al*, 2011; Gerhardt *et al*, 2013) and much stronger than that of *Nphp4-*, *Invs-* and *Nphp1-* mutant mice (Mochizuki *et al*, 1998; Morgan *et al*, 1998; Jiang *et al*, 2008, 2009; Louie *et al*, 2010; Won *et al*, 2011). In *C. elegans*, the absence of both Rpgrip1l and Cep290 disrupts the ultrastructure of the TZ (e.g. missing Y-links) (Jensen *et al*, 2015; Schouteden *et al*, 2015). In $Rpgrip1l^{-/-}$ mouse embryonic kidneys and limb buds, our TEM study confirmed the results obtained from *C. elegans rpgrip1l* mutants (Fig EV4A and B). Due to all these data and since the TZ amount of Cep290 was reduced in all analysed $Rpgrip1l^{-/-}$ mouse cell types (Figs 1D and E, and 4F and P), Cep290 might be an important factor in the development of the very severe ciliopathy of $Rpgrip1l^{-/-}$ mouse embryos. In *Drosophila melanogaster*, Rpgrip1l, Nphp4 and Nphp1 are missing and Cep290 takes over the role of Rpgrip1l as a central organiser of the TZ. In this context, Cep290 is required for the proper TZ localisation of the Mks/B9 module components (Basiri *et al*, 2014). Consistent with the data from *Drosophila*, Cep290 plays a crucial role in the assembly of the TZ in *C. elegans* where it is essential for the assembly of the Mks/B9 module but not the Nphp modules at the TZ (Li *et al*, 2016). Importantly, the localisation of Cep290 at the TZ depends on the TZ presence of Rpgrip1l in *C. elegans* (Li *et al*, 2016) and, according to the findings in *Drosophila* and *C. elegans*, there is strong suspicion that Cep290 also organises the assembly of the Mks/B9 module in vertebrates. Similar to $Cep290^{-/-}$ mouse embryos, mice having mutations in genes encoding components of the Mks/B9 module suffer from very severe ciliopathies (Reiter & Skarnes, 2006; Cook *et al*, 2009; Weatherbee *et al*, 2009; Cui *et al*, 2011; Garcia-Gonzalo *et al*, 2011; Sang *et al*, 2011; Collin *et al*, 2012). In line with the hypothesis that Rpgrip1l controls the TZ amount of the Mks/B9 module components via regulating the TZ amount of Cep290 in mice, we found a reduced TZ amount of Cep290 and all analysed Mks/B9 module components in $Rpgrip1l^{-/-}$ mouse embryonic limbs (Fig 4P–U). However, our discoveries in $Rpgrip1l^{-/-}$ MEFs and mouse embryonic kidneys contradict this hypothesis because the Mks/B9 module was properly assembled in these cells and this organ, while the amount of Cep290 was clearly decreased (Figs 1D and E, 4F–K, and EV3A–E). Consequently, it is either possible that Cep290 is required for the proper TZ amount of Mks/B9 module members in a cell type-specific way, or that the about 50% reduction of the Cep290 amount in certain tissues is sufficient to assemble the Mks/B9 module, or finally that the Cep290 function in TZ assembly is not conserved in vertebrates. To reveal the role of

Cep290 in vertebrate TZ assembly, we inactivated *Cep290* in NIH3T3 cells. In regard to the Nphp modules, the presence of Cep290 at the TZ is necessary for the proper TZ amount of Nphp1, Nphp4 and Invs (Fig 3C–E) demonstrating that Cep290 is involved in the assembly of the Nphp modules in vertebrates. Importantly, the TZ amount of Rpgrip1l is regulated independently of Cep290 (Fig 3F) placing Cep290 between Rpgrip1l and Nphp4 in the vertebrate assembly hierarchy (Fig 9A). In contrast to the Nphp modules, we could not detect any impact of Cep290 on the assembly of the Mks/B9 module (Fig EV3F–J) revealing a remarkable difference to the situation in *C. elegans*. Although the ultrastructure of the TZ in $Rpgrip1l^{-/-}$ mouse limb bud and embryonic kidney cilia was severely hampered (Fig EV4A and B), the contrast between mouse limb buds and MEFs as well as mouse embryonic kidneys in regard to the TZ amount of Mks/B9 module components (Figs 4G–K and Q–U, and EV3A–E) argues for differences in the TZ assembly hierarchies among various cell types in vertebrates. Interestingly, it was formerly shown that Rpgrip1l ensures the proper localisation of the Mks/B9 module component Tmem237 to the ciliary TZ of IMCD3 cells (Huang *et al*, 2011). This epithelial IMCD3 cell line derives from medullary collecting ducts of mice, while we studied the TZ amount of different Mks/B9 module proteins in the mesenchymal intertubular and interduct cells of mouse embryonic kidneys. These findings support our conclusion of a cell type-specific function of Rpgrip1l in the vertebrate TZ organisation, perhaps pointing towards different TZ assembly modes in mesenchymal versus epithelial tissues. To compare these data in more detail, it would be necessary to analyse Tmem237 in the intertubular and interduct cells of $Rpgrip1l^{-/-}$ mouse embryonic kidneys. If Tmem237 is not altered in the intertubular and interduct cells of $Rpgrip1l^{-/-}$ mouse embryonic kidneys, it would indicate that Rpgrip1l exerts a cell type-specific function even in the same organ. If Tmem237 is altered in these cells, the complexity of the vertebrate TZ organisation would be much higher than previously assumed. Generally, cell type-specific differences seem to be of great importance in cilia biology. For example, we detected Invs at the TZ of WT MEFs (Figs 1C and EV1C). In former studies, Invs was described as a protein which is present at the BB and/or within the cilium in IMCD3 cells, in mouse renal epithelial cells and in ciliated neurons of *C. elegans* (Shiba *et al*, 2009; Sang *et al*, 2011; Warburton-Pitt *et al*, 2012). Thus, the ciliary localisation of Invs does not only differ between different organisms but also between different cell types within the same organism. In line with this conclusion, organ-specific cilia length differences are present in $Rpgrip1l^{-/-}$ mouse embryos. While cilia in $Rpgrip1l^{-/-}$ hearts are shorter than in WT hearts, cilia in $Rpgrip1l^{-/-}$ MEFs, limbs, lungs and livers are longer than in the WT (Gerhardt *et al*, 2015). Furthermore, Mks1 regulates ciliogenesis in some but not all mouse tissues (Weatherbee *et al*, 2009) and cilia defects vary from tissue to tissue in $Tctn1^{-/-}$ mouse embryos (Garcia-Gonzalo *et al*, 2011).

Analysing the molecular mechanisms of vertebrate TZ assembling further, our data reveal that the Rpgrip1l-related and vertebrate-specific protein Rpgrip1 and Rpgrip1l synergistically control the organisation of the vertebrate TZ by ensuring the proper amount of the Mks/B9 module components at the ciliary TZ of MEFs (Figs 6E–I and 9B). Considering that Rpgrip1 arose from Rpgrip1l by gene duplication (Fig 5A), it is surprising that its localisation differs from that of Rpgrip1l. While Rpgrip1l localised to the TZ (Gerhardt

*et al*, 2015), Rpgrip1 was present at the BB (Fig 5B, C, I and J). Nevertheless, both proteins participate in the assembly of the TZ. The finding that a basal body protein ensures the proper amount of proteins at the TZ is not unique. For example, the centriolar satellite protein SSX2IP targets Cep290 to the TZ (Klinger *et al*, 2014). Moreover, Rpgrip1 directly interacts with Nphp4 and with Cep290 (Roepman *et al*, 2005; Gerner *et al*, 2010) making it plausible that Rpgrip1 is able to take part in the localisation of proteins to the TZ. Remarkably, photoreceptor cilia of $Rpgrip1^{nmf247/nmf247}$ mice lack Nphp4 (Patil *et al*, 2012). In MEFs, even Rpgrip1 alone is required for the proper TZ amount of all analysed components of the Nphp modules, although the extent of their impairment in $Rpgrip1^{nmf247/nmf247}$ MEFs is not comparable with the consequences observed in $Rpgrip1l^{-/-}$ MEFs (Figs 1A–E and 6A–D). Since the ciliary amount of Arl13b was unaltered in $Rpgrip1^{nmf247/nmf247}$ MEFs (Fig 6J), the function of the TZ as a ciliary gate does not seem to be impaired in these cells. In contrast to $Rpgrip1^{nmf247/nmf247}$ MEFs, $Rpgrip1l^{-/-}$ and $Rpgrip1l^{-/-}$; $Rpgrip1^{nmf247/nmf247}$ MEFs presented a severely decreased ciliary amount of Arl13b (Figs EV4C and 6J). In line with these molecular results, $Rpgrip1^{nmf247/nmf247}$ mouse embryos show no defects, while $Rpgrip1l^{-/-}$ mouse embryos exhibit a very severe ciliopathy (Delous *et al*, 2007; Vierkotten *et al*, 2007; Won *et al*, 2009; Gerhardt *et al*, 2013). However, the importance of Rpgrip1 in murine TZ assembling is clearly shown by the facts that the frequencies of several defects were elevated in comparison between $Rpgrip1l^{-/-}$ and $Rpgrip1l^{-/-}$; $Rpgrip1^{nmf247/nmf247}$ mouse embryos (Fig 7A) and that $Rpgrip1l^{+/-}$; $Rpgrip1^{nmf247/nmf247}$ mouse embryos display kidney cysts, while $Rpgrip1l^{+/-}$ and $Rpgrip1^{nmf247/nmf247}$ mouse embryos do not (Fig 7B and C). Since only eye defects are described in mice and humans suffering from mutations in *RPGRIP1*, most studies focussed on the relationship between Rpgrip1 and eye development. Nevertheless, it was previously suggested that Rpgrip1 is not only important for eye development but also contributes to the development of the kidneys (Roepman *et al*, 2005; Gerner *et al*, 2010). Our data confirm this hypothesis and go one step further indicating that Rpgrip1 participates at least in the development of the kidneys, the brain and the heart (Fig 7A–C).

Leaving aside the development of the eyes, the question arises why Rpgrip1 was installed in the regulation mechanism of cilia function and hence vertebrate development. It is important to ask this question because invertebrates also form functional cilia but do not have Rpgrip1 (Fig 5A) and, in addition, mutations in Rpgrip1 alone do not result in embryonic defects in mice (Zhao *et al*, 2003; Won *et al*, 2009). We propose that Rpgrip1 serves as a safeguard mechanism because, in comparison to invertebrates, the importance of functional cilia, especially of primary cilia, is far greater in vertebrates. For example, *C. elegans* and *Drosophila* have only a few cell types which are able to form cilia (Inglis *et al*, 2006; Pratt *et al*, 2016) and these two organisms develop largely properly without cilia (Apfeld & Kenyon, 1999; Basto *et al*, 2006; Bae & Barr, 2008). As our data demonstrate, Rpgrip1 is able to partially compensate for an impaired function of Rpgrip1l. Potentially, the number of ciliopathy patients would be much higher without Rpgrip1. In regard to TZ assembling, our investigations in mouse limb buds indicate that Rpgrip1 does not seem to be able to compensate the loss of Rpgrip1l in every vertebrate cell type (Fig 4M–U). This assumption is supported by the fact that the BB amount of Rpgrip1 is increased in $Rpgrip1l^{-/-}$ MEFs and mouse embryonic kidneys but not in mouse

limb buds (Fig 5E, I and J). Considering that defects in limb development are not life-threatening, one could hypothesise that Rpgrip1 exerts its function in TZ assembling exclusively in organs which are essential for survival. To test this hypothesis, further studies which compare the role of Rpgrip1 in various vertebrate cell types and organs will be necessary. Since we observed similar ultrastructural TZ defects in $Rpgrip1l^{-/-}$ limb bud and embryonic kidney cilia but could not identify the TZ in $Rpgrip1l^{-/-}$; $Rpgrip1^{nmf247/nmf247}$ embryonic kidneys (Fig EV4A and B), it is possible that additional inactivation of Rpgrip1 in some $Rpgrip1l^{-/-}$ organs might lead to the loss of the TZ. In summary, although *C. elegans* and *Drosophila* are valuable model organisms to study the basics of cilia biology, our study elucidates that in view of the development of therapies against human ciliopathies it is inevitable to analyse the mechanisms of TZ assembling in vertebrates or even mammals.

Importantly, the data obtained from $RPGRIP1L^{-/-}$ and $RPGRIP1L^{-/-}$; $RPGRIP1^{-/-}$ HEK293 cells confirmed the results quantified in MEFs and murine embryonic kidneys making it very likely that the TZ assembly organisation in humans is similar to that in mice (Figs 1A and C–E, 4C, E, F and J, 6A, C, D and H, 8A–D, and EV3D). Interesting results were shown by a recent study performed in fibroblasts obtained from patients suffering from Joubert syndrome. In these fibroblasts, mutations in *RPGRIP1L* resulted in a reduced TZ amount of the Nphp module member Nphp1 and of the Mks/B9 module components Tctn2, Ahi1 and Tmem231 (Shi *et al*, 2017). Remarkably, these data resemble those found in mouse limb buds but not those found in MEFs, mouse embryonic kidneys and HEK293 cells (Figs 1A, 4C, J, M and T, 8A and D, and EV3D). All in all, these results bring up several exciting questions: Are there any differences between TZ assembling in mice and humans? Does TZ assembling differ in various human cell types? Is TZ assembling different when comparing prenatal and postnatal conditions? To address these questions, it will be necessary to examine the TZ assembly in various (prenatal and postnatal) cell types of patients suffering from ciliopathies caused by mutations in *RPGRIP1L*. These future investigations are essential, because our data illustrate the consequences caused by a complete loss of Rpgrip1l, while most patients have hypomorphic mutations in *RPGRIP1L*. The understanding of the regulation of TZ assembling in humans is a promising research subject to pave the way for the development of therapies against ciliopathies. In this context, it should be taken into consideration that the severe ciliopathy phenotype of $Rpgrip1l^{-/-}$ mouse embryos and also the severe human ciliopathies which are caused by mutations in *RPGRIP1L* could be based on more molecular processes than the TZ assembling alone. We assume that the reduced TZ amount of Cep290 in the absence of Rpgrip1l might be very much involved in the development of the ciliopathy phenotype of $Rpgrip1l^{-/-}$ mouse embryos but since Cep290 deficiency shortened cilia in NIH3T3 cells (Fig 3G), while the loss of Rpgrip1l elongated cilia in MEFs (Gerhardt *et al*, 2015), it is unlikely that the decreased Cep290 TZ amount is solely responsible for the severe ciliopathy caused by Rpgrip1l deficiency. Based on our previous studies in mice, we reported that Rpgrip1l controls the activity of the cilia-regulated proteasome and autophagic activity (Gerhardt *et al*, 2015; Struchtrup *et al*, 2018). Importantly, Rpgrip1l regulates these degradation systems independently of each other (Struchtrup *et al*, 2018). So far, it is unknown whether the degradation systems participate in the TZ assembling or whether other TZ proteins than

Rpgrip1l govern proteasomal and/or autophagic activity. Most likely, the cilia-regulated proteasome is involved in the regulation of numerous cilia-mediated signalling pathways (Gerhardt *et al*, 2016a,b) but, as discussed by us and others before, proteasomal dysfunction is probably not the main reason for the development of ciliopathies (Liu *et al*, 2014; Gerhardt *et al*, 2015). To reveal the molecular processes underlying ciliopathies caused by mutations in *RPGRIP1L*, future studies should clarify to what extent the disturbances in TZ assembling and of the degradation systems contribute to the ciliopathy of *Rpgrip1l*$^{-/-}$ mouse embryos, whether proteasomal activity at the ciliary base and/or autophagic activity are affected in patients suffering from ciliopathies caused by mutations in *RPGRIP1L* and if there are interactions between TZ assembling and the degradation systems which ensure the proper development and homeostasis in vertebrates.

# Materials and Methods

### Animal husbandry and ethics statement

All mice (*Mus musculus*) used in this study were on the C3H background (*Rpgrip1l* and *Rpgrip1* mutant mice) or on C57BL/6J background (*Nphp4* mutant mice) and kept under standard housing conditions with a 12/12-h dark–light cycle and with food and water *ad libitum*. All animal procedures were carried out in accordance with National Institutes of Health guidelines and with local and state regulations for research with animals.

### Mouse strains

The generation of *Rpgrip1l* and *Rpgrip1* mutant mice and their genotyping has been delineated previously (Vierkotten *et al*, 2007; Won *et al*, 2009). *Nphp4* mutant mice were generated by inserting *loxP* sites within the introns 13 and 16 (in collaboration with ICS, Strasbourg). *Nphp4*$^{flox/flox}$ mice were obtained by DNA recombination, inserting LoxP sites flanking the 5′ and 3′ regions of the exons 14–16 in the *Nphp4* gene in embryonic stem cells line 129/Sv using a *PGK-neo* cassette as a selectable marker. These mice were mated with a Cre transgenic mouse line to obtain *Nphp4*$^{-/-}$ mice with genomic deletion of the exons 14–16. Genotypes were determined with PCR amplification of total genomic DNA, using 2 pairs of primers: upper primer in intron 13: 5′-CTG AGA CAG GAC AGG AAC GTG ATG C-3′ and lower in intron 13/exon 14: 5′-CTG CTC TCT AAT ACA GAT TCC TGC C-3′ (WT strand) or in intron 17: 5′-TCA CAG TCA CAG TGA CTC CAA GTC C-3′ (deleted strand). The genomic deletion of exon 14–exon 16 of the *Nphp4* gene is predicted to lead to frame-shift.

### Antibodies and plasmids

We used primary antibodies to actin (A2066; Sigma-Aldrich), Arl13b (17711-1-AP; Proteintech), Cep290 (ab84870; Abcam), Gapdh (G8795; Sigma-Aldrich), Ift88 (13967-1-AP; Proteintech), Invs (10585-1-AP; Proteintech), HA (sc-805; Santa Cruz Biotechnology, Inc.), Mks1 (16206-1-AP; Proteintech), Myc (sc-789; Santa Cruz Biotechnology, Inc.), Nphp1 (sc-20204; Santa Cruz Biotechnology, Inc.), Nphp4 (sc-49246; Santa Cruz Biotechnology, Inc.), Rpgrip1

(13214-1-AP; Proteintech), Tctn1 (15004-1-AP; Proteintech), Tctn2 (17053-1-AP; Proteintech), Tctn3 (16085-1-AP; Proteintech), Tmem67 (13975-1-AP; Proteintech), acetylated α-tubulin (sc-23950; Santa Cruz Biotechnology, Inc.), detyrosinated α-tubulin (AB3201; EMD Millipore), γ-tubulin (T6557; Sigma-Aldrich) and γ-tubulin (sc-7396; Santa Cruz Biotechnology, Inc.). The generation of the polyclonal antibody against Rpgrip1l was described formerly (Vierkotten *et al*, 2007). Antibodies were affinity-purified with the antigen coupled to Ni-NTA agarose (Qiagen #30230). Immunofluorescence analyses were performed to confirm the specificities of the antibodies (Appendix Fig S13). Antibody specificities of anti-Rpgrip1, anti-Invs, anti-Mks1, anti-Tmem67, anti-Tctn2 and anti-Tctn3 antibodies have been shown by using appropriate antigens [recombinant Rpgrip1 (ag4089; Proteintech), recombinant Invs (ag17782; Proteintech), recombinant Mks1 (ag9177; Proteintech), recombinant Tmem67 (ag5174; Proteintech), recombinant Tctn2 (ag10725; Proteintech), recombinant Tctn3 (ag9106; Proteintech)].

The following plasmids were used: pRpgrip1l, pNphp4 (kindly provided by Ronald Roepman), pNphp1, pInvs (Addgene #54131) and IFT88-EYFP (kindly provided by David R. Beier). pRpgrip1l encodes for the human full-length Rpgrip1l protein fused to a myc-tag (vector: pRRLsin), pNphp4 encodes for the murine full-length Nphp4 protein fused to a HA-tag (vector: pcDNA3), pNphp1 encodes for the murine full-length Nphp1 protein fused to a myc-tag (vector: CMV) and pInvs encodes for the murine full-length Invs protein fused to mEmerald (vector: mEmerald-C1). The cilium in the *in situ* PLA experiments was marked by the product of a transiently transfected IFT88-EYFP construct (Fig 5D and Appendix Fig S11A and B). The IFT88-EYFP fusion protein was present in the pJAG368 vector and expressed under the control of the SV40 promoter.

### Cell culture and transfection

Mouse embryonic fibroblasts were isolated from single mouse embryos after standard procedures. MEFs, NIH3T3 (mouse embryonic fibroblast cell line) and HEK293 (human embryonic kidney) cells were grown in DMEM supplemented with 10% foetal calf serum (FCS), 1/100 (v/v) L-glutamine (Gibco), 1/100 (v/v) sodium pyruvate (Gibco), 1/100 (v/v) non-essential amino acids (Gibco) and 1/100 (v/v) pen/strep (Gibco) at 37°C and 5% $CO_2$. For induction of ciliogenesis, MEFs and NIH3T3 were grown to confluency and serum-starved with medium containing 0.5% FCS for at least 24 h. For DNA transfection, Lipofectamine 3000 (Invitrogen) was used following the manufacturers' guidelines.

### CRISPR/Cas9-mediated gene inactivation

See Appendix Supplementary Methods.

### Cystic index calculation

The extent of cyst formation was quantified in transversal sections of whole embryonic kidneys (E18.5). Six sections were investigated for each experimental animal. Whole kidney images were obtained using the Axioskop 2 system (with an AxioCamMRc camera and the AxioVision 4.7.1 software; Carl Zeiss AG). Total kidney area and total cystic area were quantified by using Fiji

(National Institutes of Health) in batch mode applying a manual threshold of 230. Cystic index = (total cystic area/total kidney area) × 100.

## Histology

Embryos were dissected and fixed in 4% paraformaldehyde (PFA) overnight at 4°C. Subsequently, they were serially dehydrated using ethanol, embedded in paraffin and sectioned (12 μm). The sections were stained with haematoxylin and eosin.

## Image processing

Image acquisition of single plane images and data analysis were carried out at room temperature using a Zeiss Imager.A2 microscope, 100×, NA 1.46 oil immersion objective lens (Carl Zeiss AG), a monochrome charge-coupled device camera (AxioCam MRm, Carl Zeiss AG), and the AxioVision Rel. 4.8 software package (Carl Zeiss AG).

3D-structured illumination microscopy (3D-SIM) was performed on a Zeiss ELYRA PS.1 system (Carl Zeiss Microscopy GmbH, Göttingen, Germany) equipped with an Andor EM-CCD iXON DU-885 with 1,004 × 1,002 pixels at room temperature. Z-stacks were taken by using a 100× alpha-Plan-Apochromat oil immersion objective with a numerical aperture of 1.46. To generate structured illumination, a grid pattern is projected onto the image plane in five different positions and at five different modulation angles to obtain high-frequency information within the low-frequency information captured by the optical system. For the Dylight405-, Dylight488-, Cy3-channel back-computation of the lower frequencies using Fourier transformation was performed using the Zeiss ZEN Structured Illumination Processing tool to increase the resolution in the final image. Dylight405, Dylight488, Cy3, Alexa 405 and Alexa 488 were used as fluorochromes.

## Immunofluorescence

For immunofluorescence on MEFs, NIH3T3 cells and HEK293 cells, cells were plated on coverslips, and after reaching confluency, they were serum-starved for at least 24 h. Cells were fixed with 4% PFA (for stainings with the antibodies to Rpgrip1l, Nphp4, Invs, Cep290 and Arl13b) or methanol (for stainings with the antibodies to Rpgrip1, Nphp1, Mks1, Tmem67, Tctn1, Tctn2 and Tctn3). Fixed cells were rinsed three times with PBS, following permeabilisation with PBS/0.5% Triton X-100 for 10 min. After three washes with PBS, they were blocked for at least 10 min at room temperature in PBST (PBS/0.1% Triton X-100) containing 10% donkey serum. Diluted primary antibodies in block were incubated overnight at 4°C. After three washing steps with PBST, incubation with fluorescent secondary antibody diluted in block was performed at room temperature for 1 h followed by several washings and subsequent embedding with Mowiol optionally containing DAPI (Merck #1.24653).

For immunofluorescence on cryosections, embryos were fixed in 4% PFA and incubated in 30% sucrose (in PBS) overnight at 4°C. Then, they were embedded in Tissue-Tek O.C.T. compound (Sakura Finetechnical #4583) and stored at −80°C. Transverse cryostat sections (7 μm in thickness) were prepared, washed with PBS and permeabilised with PBS/0.5% Triton X-100. Blocking was performed with 10% donkey serum in PBST. The sections then underwent an incubation in the primary antibodies diluted in block overnight at 4°C. After three washing steps, they were incubated with the secondary antibody (diluted in block) for 2 h and were washed again. Finally, they were embedded in Mowiol.

## In situ PLA

The in situ proximity ligation assay (PLA or DUOLINK™) was performed according to the manufacturer's manual (OLink Bioscience) and was used to estimate the distance between Rpgrip1l and Rpgrip1 at the ciliary base. Firstly, Rpgrip1l and Rpgrip1 are recognised by specific primary antibodies. Subsequently, PLA probes consisting of species-specific secondary antibodies, each with a unique short DNA strand attached to it, bind to the primary antibodies. The addition of two other circle-forming DNA oligonucleotides leads to an interaction of the DNA strands, if Rpgrip1l and Rpgrip1 are in close proximity (< 40 nm). After joining of the two added oligonucleotides by enzymatic ligation, a rolling circle amplification using a polymerase starts. The amplified DNA is made visible by fluorescent-labelled complementary oligonucleotide probes.

## Off-target analysis

See Appendix Supplementary Methods.

## Quantifications

Intensity of ciliary protein staining on single plane images and of protein bands was quantified by using ImageJ (National Institutes of Health). Intensity of ciliary proteins based on immunofluorescence stainings was measured as described before (Garcia-Gonzalo et al, 2011, 2015; Gerhardt et al, 2015; Roberson et al, 2015; Yee et al, 2015; Struchtrup et al, 2018). For the quantification of Arl13b, we used the area marked by acetylated α-tubulin and quantified the average pixel intensity to take the cilia length into account thereby making the data of WT and $Rpgrip1l^{-/-}$ cilia comparable. For all other ciliary protein intensities, we selected the region labelled by γ-tubulin (for Rpgrip1) or the area in-between the γ-tubulin staining and the proximal part of the acetylated α-tubulin staining (for TZ proteins) and measured the total pixel intensity. The labelling of the selected region was outlined with the freehand selection tool and the mean intensity of the desired channel inside the area measured in an 8-bit scale (0-255). To get rid of the ratio of unspecific (background) staining, we subtracted the mean value of the average pixel intensity (in the case of Arl13b) or of the total pixel intensity (for Rpgrip1 and TZ proteins) of three neighbouring regions free from specific staining. The quantified areas of the specific staining and the unspecific staining of every individual measurement have to be equal in size. The size of the quantified region is determined by the size of the area in-between the γ-tubulin staining and the proximal part of the acetylated α-tubulin staining when the specific signal is quantified.

Intensity quantification of the Cep290 fluorescence signal which was obtained by using 3D-SIM was performed as follows: Surface

reconstructions of Cep290 were performed using Imaris 8.3 (Bitplane). 3D-SIM images of WT and *Rpgrip1l*$^{-/-}$ MEFs were analysed in batch mode applying a manual threshold of 13,000 and a surface area detail of 50 nm. For each image, we manually selected the surface of CEP290 between the BB (γ-tubulin) and the ciliary axoneme (acetylated α-tubulin) to measure the summarised intensity. If necessary, two signals were separated manually using the cut surface function of Imaris. For images where no surface was detected in the TZ, measurements were set to 0.

### Real-time PCR analysis

Mouse embryonic fibroblast RNA was isolated by using RNeasy Kit (Qiagen #74104) and RNase-Free DNase Set (Qiagen #79254). Isolated RNA was converted into cDNA by utilising Expand Reverse Transcriptase (Roche #11785826001). Quantitative real-time PCR was performed by employing a Step One Real-Time PCR System Thermal Cycling Block (Applied Biosystems #4376357) and the TaqMan Universal PCR Master Mix, No AmpErase UNG (Applied Biosystems #4324020). The following primer/TaqMan probe sets were used: *Hprt* (Assay ID: Mm01545399_m1) and *Nphp4* (Assay ID: Mm00463212_m1). Real-time PCR was carried out with 50 ng of MEF cDNA of each sample in triplicate reactions in a 20 μl volume containing 100 nM primers and 50 nM probe. Cycling conditions were 50°C for 2 min and 95°C for 10 min, followed by a 40-cycle amplification of 95°C for 15 s and 60°C for 1 min. The analysis of real-time data was performed by using included StepOne Software version 2.0.

### Statistical data

Data are presented as mean ± standard error of mean (SEM). Two-tailed Student's *t*-test was performed for all data in which two datasets were compared. Analysis of variance (ANOVA) and Tukey honest significance difference (HSD) tests were used for all data in which more than two datasets were compared. Differences in defect frequencies were examined via chi-square tests. A *P*-value < 0.05 was considered to be statistically significant (one asterisk), a *P*-value < 0.01 was regarded as statistically very significant (two asterisks), and a *P*-value < 0.001 was accounted statistically high significant (three asterisks).

### Transmission electron microscopy

E12.5 and E18.5 embryos were isolated in PBS, and the limbs (E12.5) and the kidneys (E18.5) were dissected and fixed in 2.5% glutaraldehyde. Samples were washed in six changes of 0.1 M cacodylate buffer for 60 min, post-fixed in 2% OsO$_4$ for 120 min at room temperature, washed again in 6 changes of 0.1 M cacodylate buffer for 60 min, and dehydrated through a graded ethanol series. The samples were infiltrated with propylene oxide and embedded in Araldite (Serva) allowing to polymerise at 60°C for 2 days. After trimming the specimen, semi-thin sections (1 μm) were made and stained with Richardson solution for choosing the area of interest. Ultrathin sections (70 nm) were cut with a diamond knife using an ultramicrotome (Ultracut E, Reichert). The cuts were placed on 75-mesh copper grids (Plano), and finally, the slides were contrasted with uranyl acetate and lead

citrate. EM images were acquired by using Zeiss A902 (Carl Zeiss AG, Oberkochen, Germany) at an accelerated voltage of 80 kV or by using H600 TEM (Hitachi, Tokyo, Japan) at 75 kV using DigitalMicrograph connected to a MSC TK1024M (Gatan, Munich, Germany).

### Western blotting

Western blot studies were done essentially as described using anti-Invs, anti-Cep290, anti-Rpgrip1, anti-Rpgrip1l, anti-Myc, anti-Nphp4, anti-HA and anti-Nphp1 antibody (Wang *et al*, 2000). Anti-actin or anti-Gapdh antibody was used as loading control. Visualisation of protein bands was realised by LAS-4000 mini (Fujifilm). Band intensities were measured by using ImageJ (National Institutes of Health).

**Expanded View** for this article is available online.

### Acknowledgements

The authors thank the staff of the Center for Advanced imaging (CAi) at the Heinrich Heine University Düsseldorf for their support concerning the transmission electron microscopy and the three-dimensional structured illumination microscopy—in particular Marion Nissen and Sebastian Hänsch. Additionally, we would like to express our thanks to Elisabeth Wesbuer and Andreas Barbian for their short-term and straightforward aid in terms of transmission electron microscopy. Moreover, we are grateful to Karl W. Kafitz and Niklas J. Gerkau for their assistance regarding the analysis of microscopic images and to Stephan Burmühl for his help in terms of electron micrographs. We are indebted to Patsy Nishina for providing mice carrying the *Rpgrip1*$^{nmf247}$ allele. Special thanks go to David R. Beier for providing the Ift88-EYFP construct and to Ronald Roepman for providing the Nphp4-HA construct. Furthermore, we thank Katharina Everaerts, Julia Perrar and Oliver Koch for their participation in the off-target analysis of the cell clones which were generated by using the CRISPR/Cas9 system. This work was supported by the Deutsche Forschungsgemeinschaft (Sonderforschungsbereiche 590 and 612) to U.R. The project of the S.S.M. laboratory was funded by the Agence Nationale pour la Recherche (ANR blanc project 11-BSV2-0006 to S.S.M.) and the Fondation pour la Recherche Médicale (Equipe FRM DEQ20140329544 to S.S.M.).

### Author contributions

CG conceived the study and designed experiments. AW, LK, SK, JML and TL performed immunofluorescence studies with subsequent intensity quantifications, cilia length measurements and statistics. AW performed all histological studies (with assistance from SK and TL). AW measured defect frequencies. AW performed the transmission electron microscopic studies. RD, LK and SD inactivated several genes in different cells via the CRISPR/Cas9 system. ASp, SK and CG performed the Western blot investigations. RD performed phylogenetic and evolutionary studies. TZ and CG performed fluorescence intensity quantifications of 3D reconstructed images obtained by using 3D-SIM. ASt performed the qRT–PCR analysis. FL, CV and SS-M generated *Nphp4*$^{-/-}$ MEFs. FL and SS provided *Nphp4*$^{-/-}$ mouse embryos. AW and RD prepared figures. CG wrote the manuscript with assistance from AW and UR. All authors approved the manuscript. UR obtained funding for the study.

### Conflict of interest

The authors declare that they have no conflict of interest.

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
