## [Review Process File · The EMBO Journal]

Cell type-specific regulation of ciliary transition zone assembly in vertebrates

Antonia Wiegeling, Renate Dildrop, Lisa Kalfhues, André Sychala, Stefanie Kuschel, Johanna Maria Lier, Thomas Zobel, Stefanie Dahmen, Tristan Leu, Andreas Struchtrup, Flora Legendre, Christine Vesque, Sylvie Schneider-Maunoury, Sophie Saunier, Ulrich R  ther and Christoph Gerhardt

Review timeline:

Submission date:	14 July 2017
Editorial Decision:	10 August 2017
Additional correspondence	11 October 2017
Revision received:	7 February 2018
Editorial Decision:	1 March 2018
Revision received:	12 March 2018
Accepted:	15 March 2018

Editor: Ieva Gailite

Transaction Report:

1st Editorial Decision

10 August 2017

Thank you for submitting your manuscript for consideration by the EMBO Journal. We have now received three referee reports on your manuscript, which I have included below for your information.

As you can see from the comments, reviewers #1 and #2 appreciate the significance of the presented characterisation of ciliary transition zone assembly in mammals. However, the reviewers also raise several concerns that would have to be addressed before they can support publication of the manuscript. I would like to invite you to submit a revised version of the manuscript while addressing the comments of all reviewers, but particularly focusing on the following points:

1. Analysis of the effects of Rpgr1/Rpgr1L loss on transition zone architecture using super-resolution microscopy or electron microscopy (Reviewer #2, point 6 and Reviewer #3, point 3).
2. Improve quantification of colocalisation analysis as requested by reviewer #3, point 1.
3. Extend the characterisation of used mutant alleles as requested by referee #1 (point 3) and referee #2 (points 2 and 4).

When preparing your letter of response to the referees' comments, please bear in mind that this will form part of the Review Process File, and will therefore be available online to the community. For more details on our Transparent Editorial Process, please visit our website: http://emboj.embopress.org/about#Transparent_Process

We generally allow three months as standard revision time, but an extension to six months is possible in the case of substantial revisions. Please contact us in advance if you would need an additional extension. As a matter of policy, competing manuscripts published during this period will not negatively impact on our assessment of the conceptual advance presented by your study.

However, we request that you contact the editor as soon as possible upon publication of any related work to discuss how to proceed.

Please feel free to contact me if have any further questions regarding the revision. Thank you for the opportunity to consider your work for publication. I look forward to your revision.

Referee #1:

Previous work, most notably in *C. elegans*, has investigated the hierarchy by which various ciliopathy proteins are assembled at the TZ. The current model from worm work is that RPGRIP1L is the key regulator by which all of these proteins incorporate into the TZ. However, this model has not been rigorously tested in vertebrates, which also possess an RPGRIP1L paralogue (RPGRIP1) that is missing in *C. elegans* and other invertebrates. The overall premise of the study is important because of the very strong TZ-ciliopathy association, and because a hierarchical TZ assembly model provides a means for understanding the variation in phenotype associated with patients with mutations in corresponding TZ genes.

Wiegeling et al rigorously tests the prevailing assembly model using loss of function or null alleles of various mouse and human TZ genes. In MEFs, mouse kidney and HEK293 cells, RPGRIP1L alone directs TZ incorporation of NPHP module proteins, whereas the TZ incorporation of MKS/B9 module proteins requires RPGRIP1L and RPGRIP1 functioning in a redundant manner. However, in mouse embryonic limbs, RPGRIP1L alone directs the TZ incorporation of NPHP and MKS/B9 module proteins. These results agree with the prevailing model that RPGRIP1 genes (RPGRIP1 and 1L) are at the apex of a hierarchical TZ assembly pathway; importantly, the findings also uncover novel distinct cell type-specific mechanisms by which RPGRIP1/L regulates TZ assembly in mammals.

Overall, the work is well executed and convincing, with the important conclusions supported by the data. There is currently major interest in the TZ, both from a ciliopathy perspective, and because the TZ is now a paradigm for ciliary gating and the biology of diffusion barriers. Thus, the study should be of interest to a wide audience.

Minor concerns:

1. Whilst the manuscript is generally easy to follow, the writing could be improved in places. There are some grammatical issues, and the discussion text is sometimes a little long-winded and repetitive.
2. Not all relevant papers on TZ assembly hierarchy are cited. Schouteden et al; Lambacher et al.; Craige et al. 2010
3. The authors must make clear what type of alleles they are working with in the various mice and cell lines, which I assume are nulls or at least very severe loss of function. Also, it is essential to provide the details of the specific mutations they generated in the CRISPR/Cas9-disrupted cells. Also, the authors should show western blots for the corresponding proteins in the k/o cells - whilst IF shows that the TZ signals for the corresponding proteins are gone in the k/o cells, this is not the same as showing that the protein is gone from the cell.
4. The finding that Inv localizes at the TZ is surprising since a number of studies in mammalian cells and worms specifically showing that this protein is not at the TZ. I think this deserves some further discussion in the manuscript.
5. References are missing from the second last sentence on Page 14.
6. On page 15, the authors discuss why MKS/B9 levels do not change in RPGRIP1L^{-/-} MEF/kidney cilia, despite a reduction in CEP290 TZ signals. To test some of the reasons they provide for this observation, one could profile the TZ levels of MKS/B9 proteins against the levels of CEP290 RPGRIP1L^{-/-} cilia costained for both proteins? For example, in more extreme examples where

CEP290 levels are very low (80-90% reduced), do you still see normal levels of MKS/B9 proteins?

7. N-values not provided for data in FigS4

8. In Fig. S3, the ARL13 signal looks like it is in distal regions of RPGRIP1L $-/-$ cilia. Is this a representative image, and if so, is there any significance to this observation?

9. Although the IF images indicate that RPGRIP1L and RPGRIP1 are expressed in all examined cells, some RNA analysis (q-PCR or similar) to support this would be useful.

10. Details on the antigens used in FigS5 are missing.

Referee #2:

In this manuscript Wiegering et al. investigate the role of Rpgrip11, Nphp4 and Rpgrip1 in regulating the localization of ciliary transition zone proteins. Primary cilia are sensory organelles present on many cell types. Mutations of genes encoding for ciliary proteins can lead to a variety of diseases referred to as ciliopathies. The most prominent and frequent form is Cystic Kidney Disease, while the actual number of different ciliopathies is still increasing. All ciliopathies lack any causative therapy. Therefore, a detailed understanding of cilia function and especially the function of ciliopathy genes is urgently required. The at hand study focuses on the transition zone (TZ), which is localized between the basal body and the ciliary shaft. Here, the entry and exit in cilia is regulated and many disease genes encode to TZ proteins. The authors primarily focus on MKS5 alias Rpgrip11 or NPHP8 and aim to confirm data from *C. elegans* on the important role of Rpgrip11 at the TZ. The authors show, that in murine tissues and cultured cells (and in human cells) Rpgrip11 regulates the localization of Nphp4, Nphp2 (*invs*), Nphp1, Nphp6 (Cep290) and other TZ players, to different extents in different cell types/tissues. Here, Rpgrip1 as synergistic partner of Rpgrip11 might be important.

In summary, this is an interesting study on an interesting cell biological and biomedical topic. The study uses extensive ciliary IF imaging and is to this end very carefully performed. However, the study remains primarily descriptive and does not gain many new mechanistic insights into the regulation of the TZ proteins that are investigated. It does not really address the question of how Rpgrip1L and Rpgrip1 actually do ensure TZ assembly. On the plus side, however, most observation were made in vivo using the appropriate genetic mouse models or at least Mef's derived from those. Although I really like this systematic and thoroughly performed imaging study, I have a number of concerns that should be addressed by the authors:

1) Key finding of Fig. 1 and Fig. 2 is the reduction of Nphp1/4 or Nphp1/*Invs* at the TZ in Rpgrip11 $-/-$ or in NPHP4 $-/-$ MEFs, respectively. This is not entirely convincing. Here, the authors should explain more detailed the provenience of the MEFs. Are they derived from littermates sharing exactly the same genetic background? Have the results been confirmed in independent MEF cell lines to exclude any (clonal) cell culture artefacts? To ultimately prove the specificity of their findings Rpgrip11 or Nphp4 should be re-expressed in the respective KO MEFs. An even partial rescue in these easy experiments would underline the specificity of these data.

2) The authors use Crispr/Cas9 to generate cell lines lacking expression of Nphp1 and Inversin (Fig. 2 J,K). The authors should provide the genetic information of the cell lines presented. What is the nature of the on-target mutation? How did they exclude off-target effects? Either additional clones based on different sgRNAs could be used or rescue experiments should be performed to demonstrate specificity. Moreover, the knockout should be proven by western blot and not only by IF stainings of the GOI at the TZ.

3) In Fig. 6 the authors switch to HEK cells as they state "to get an idea if these data obtained from mice are transferable to humans". There are indeed a view reports on primary cilia in HEK cells. However, HEK cells are not an ideal model to study cilia biology or human diseases. Therefore, the authors might rather use more differentiated ciliated human cell lines (e.g. RPE (retina), HK2 (kidney), ...).

4) As mentioned above for the Nih3t3 Crispr/Cas9 cells, the knockout RPGRIP1L and RPGRIP1 in HEK cells must be proven by western blot. What is the genotype of the cell clones? Additional

clones should be added and/or re-expression experiments performed to show specificity.

5) Interestingly, heterozygous *Rpgrip1l* mice combined with the *Rpgrip1 nmf247* allele develop polycystic kidneys underlining their synergism *in vivo*. I feel that this important phenotypic finding should not be "hidden" in Fig S4 but presented in more detail e.g. in Fig. 5.

6) Most of the data presented in this study demonstrates quantitatively altered localization of Nphp proteins at the TZ. This raises the question whether the loss of the *Rpgrip1L/Rpgrip1* alters the architecture and ultrastructure of the transition zone? This could be studied by some more super resolution imaging (such as 3D-Sim) but ultimately might require electron microscopic imaging of the TZ in the *Rpgrip1* mutant cells.

Minor points:

- pg 19, methods: Regarding the *Nphp4* mice: Why (and how) was cre recombination done in ES cells?

- The legend of Fig. 5 L does not mention the correct genotypes "(L) Illustration of several defect frequencies in WT, *Rpgrip1nmf247/nmf247* and *Rpgrip1nmf247/nmf247*; *Rpgrip1nmf247/nmf247* mouse embryos".

- The authors should provide more details and comments on the image acquisition and quantification of TZ proteins.

Referee #3:

The manuscript by Wiegeling et al describes extensive *in vitro* and *in vivo* cell biology studies that delineate the relative roles of the related proteins RPGRIP1L and RPGRIP1 in maintaining the ciliary transition zone. The important findings from this study are that RPGRIP1L and RPGRIP1 act synergistically to maintain the correct localization of transition zone proteins, and that RPGRIP1L contributes to this process in a cell-type specific manner.

The authors base these conclusions on quantitative immunofluorescence studies using an extensive panel of mouse embryonic fibroblasts, kidneys and limb buds derived from *Rpgrip1l*, *Rpgrip1* and *Nphp4* knock-out mice. Double knock-out lines in HEK293 cells have also been generated and characterized. However, in all of these experiments, the "total pixel intensity" method of quantitation is unclear to me. In each of the bar graphs, I assume that the black bars are intensity values from wild-type cells normalized to 100% and grey bars are mutant cells. Comparisons using total pixel intensity could then be easily confounded by technical issues and variations in staining efficiency. For me, a more robust method would be a measure of co-localization within each experiment using a 3D reconstruction from a confocal z-stack. This is reasonably straight-forward in software such as Velocity, and possible with more effort in FIJI. A more subtle effect could be possible from differences in cell cycle progression between wild-type and mutant cell-lines, or differences in the relative rates of either ciliogenesis or transition zone assembly. To exclude the latter possibility, live cell imaging could be done in the MEF or knock-out NIH3T3 or HEK293 cell-lines. The authors' work suggests that CEP290 localization would be a good one to test.

The work also suggests that RPGRIP1 is able to compensate, at least partially, for RPGRIP1L in MEFs (Figure 5). Since the phenotypic effects of human RPGRIP1 mutations are limited to the photoreceptors (causing Leber congenital amaurosis), it would therefore also be interesting to test the extent of RPGRIP1L/RPGRIP1 synergy in the photoreceptor connecting cilium. One might predict that RPGRIP1 would not be able to compensate for RPGRIP1L because the retinal-specific function of RPGRIP1.

However, what I miss in these experiments is a more detailed consideration of the possible mechanistic or structural basis of RPGRIP1L/RPGRIP1 synergy. RPGRIP1L and RPGRIP1 appear to have different localizations to the transition zone (Figure 2A) and basal body (Figure 4C-E), respectively, at least in MEFs. (This is also shown in Figure 4B). Could super resolution microscopy allow better characterization of this localization and ask if other transition zone proteins are mis-localized? For example, CEP290 appears to be mis-localized in *Nphp4* knock-out MEFs compared to wild-type (Figure 2D).

Finally, the phrase "mice and men" is used a couple of times, and although I also appreciate the

poetry of Robert Burns, I think that it is inappropriate in this context because it could be misconstrued as sexist.

Additional correspondence (author)

11 October 2017

We are currently revising our work by taking the comments of the referees as a guide for the correction and quality improvement of our manuscript. Since we attach the greatest importance to meet the requirements of the three referees, the processing time of 3 months is too short. For the following reasons, we kindly request you to extend the processing time from three to six months:

1) Referee #1 requested (point 6): On page 15, the authors discuss why MKS/B9 levels do not change in Rpgrip1L^{-/-} MEF/kidney cilia, despite a reduction in CEP290 TZ signals. To test some of the reasons they provide for this observation, one could profile the TZ levels of MKS/B9 proteins against the levels of CEP290 Rpgrip1L^{-/-} cilia costained for both proteins? For example, in more extreme examples where CEP290 levels are very low (80-90% reduced), do you still see normal levels of MKS/B9 proteins?

We think that this question is one of the open key questions within our manuscript. However, the kind of measurement suggested by referee #1 is very variable because it would be made under poorly defined conditions. To get a more defined condition than to look for very low levels of Cep290 in Rpgrip11^{-/-} MEFs and embryonic kidneys, we are currently generating Cep290^{-/-} NIH3T3 cells (immortal MEF cell line) and will quantify not only the Mks/B9 module members but also Rpgrip11 and the Nphp module members Nphp1, Invs and Nphp4 in these cells. In this way, we will be able to analyse the role of Cep290 in ensuring the amount of the Mks/B9 module members in MEFs/NIH3T3 cells and, additionally, to determine the position of Cep290 within the Nphp module. Certainly, the generation of these cells, the confirmation of the inactivation of Cep290 in these cells (via Western blot studies, immunofluorescence stainings and rescue experiments) and the quantification of the amount of the above-mentioned TZ proteins cannot be done in three months but we are able to manage these analyses in six months.

2) Referee #2 wrote (point 6): Most of the data presented in this study demonstrate quantitatively altered localization of Nphp proteins at the TZ. This raises the question whether the loss of the Rpgrip1L/Rpgrip1 alters the architecture and ultrastructure of the transition zone? This could be studied by some more super resolution imaging (such as 3D-SIM) but ultimately might require electron microscopic imaging of the TZ in the Rpgrip1 mutant cells.

Since it is impossible to analyse the ultrastructure of the transition zone (TZ) in detail by using 3D-SIM, we decided to use transmission electron microscopy (TEM). Our aim is to compare the ultrastructure of the transition zone between the wild-type and Rpgrip11-negative status but also the TZ ultrastructure between cilia of Rpgrip11^{-/-} embryonic limbs and embryonic kidneys as well as the TZ ultrastructure between wild-type, Rpgrip11^{-/-} and Rpgrip11^{-/-}; Rpgrip1 nmf247/nmf247-combined embryonic limbs and kidneys. Due to the wealth of TEM examinations, we are not able to finish the TEM analyses until 8th November 2017.

3) All in all, the reviewers recommended a plethora of investigations

and besides that some of them are time-consuming since the end of their completion is not easily predictable. For example, we are currently working on the confirmation of the particular gene inactivation in the mutant cells by Western blot analyses and re-expression of the inactivated genes (rescue experiments). In case of the re-expression studies, we already made the expression constructs ready to use but we do not finished all transfections until now. In particular the transfection into MEFs is time-consuming because primary cells are difficult to transfect so that we need several attempts. Since this is only the prerequisite to perform immunofluorescence stainings for transition zone proteins in order to show a rescue, we cannot complete all these investigation on time.

Additional correspondence (editor)

11 October 2017

Thank you for getting in touch regarding the revision timeline for your manuscript. I understand that the extensive experiments you are undertaking would need a longer revision period, so I am happy to extend the revision deadline to six months. We will adjust the deadline in our system accordingly.

1st Revision - authors' response

7 February 2018

Referee #1:

*Previous work, most notably in *C. elegans*, has investigated the hierarchy by which various ciliopathy proteins are assembled at the TZ. The current model from worm work is that RPGRIP1L is the key regulator by which all of these proteins incorporate into the TZ. However, this model has not been rigorously tested in vertebrates, which also possess an RPGRIP1L paralogue (RPGRIP1) that is missing in *C. elegans* and other invertebrates. The overall premise of the study is important because of the very strong TZ-ciliopathy association, and because a hierarchical TZ assembly model provides a means for understanding the variation in phenotype associated with patients with mutations in corresponding TZ genes.*

Wiegering et al rigorously tests the prevailing assembly model using loss of function or null alleles of various mouse and human TZ genes. In MEFs, mouse kidney and HEK293 cells, RPGRIP1L alone directs TZ incorporation of NPHP module proteins, whereas the TZ incorporation of MKS/B9 module proteins requires RPGRIP1L and RPGRIP1 functioning in a redundant manner. However, in mouse embryonic limbs, RPGRIP1L alone directs the TZ incorporation of NPHP and MKS/B9 module proteins. These results agree with the prevailing model that RPGRIP1 genes (RPGRIP1 and 1L) are at the apex of a hierarchical TZ assembly pathway; importantly, the findings also uncover novel distinct cell type-specific mechanisms by which RPGRIP1/L regulates TZ assembly in mammals.

Overall, the work is well executed and convincing, with the important conclusions supported by the data. There is currently major interest in the TZ, both from a ciliopathy perspective, and because the TZ is now a paradigm for ciliary gating and the biology of diffusion barriers. Thus, the study should be of interest to a wide audience.

Minor concerns:

1. Whilst the manuscript is generally easy to follow, the writing could be improved in places. There are some grammatical issues, and the discussion text is sometimes a little long-winded and repetitive.

We revised the text very carefully and avoided repetitions. The discussion part was re-organised and shortened. However, some new data were discussed and hence the total length of the discussion part did not alter noticeably.

2. Not all relevant papers on TZ assembly hierarchy are cited. Schouteden et al.; Lambacher et al.; Craige et al. 2010

We thank Referee #1 for pointing that out. The mentioned articles have been inserted into the manuscript (p. 4, 11 and 17). Due to the new results we present in the revised version of the manuscript, we added several other reports. Moreover, we discussed our data in the context of the recently published results of Shi et al and inserted the relevant reference into the manuscript (Shi et al, 2017) (p. 11 and 23).

3. The authors must make clear what type of alleles they are working with in the various mice and cell lines, which I assume are nulls or at least very severe loss of function. Also, it is essential to provide the details of the specific mutations they generated in the CRISPR/Cas9-disrupted cells. Also, the authors should show western blots for the corresponding proteins in the k/o cells - whilst IF shows that the TZ signals for the corresponding proteins are gone in the k/o cells, this is not the same as showing that the protein is gone from the cell.

We added more information about the clones which were generated by using the CRISPR/Cas9 system in the revised version of the manuscript including the genotype analyses (Appendix Fig S3). Moreover, we performed Western Blot studies which confirm the absence of these proteins in the entire cells (Appendix Figs S2E, S10A-C, S12A and B). However, we did not carry out Western Blot studies for the *Rpgrip11* mutant mice and MEFs (which were obtained from these mice) used in this study as well as for the *Rpgrip1^{nmf247}* mutant mice and MEFs (which were obtained from these mice) used in this study, since these data have been published before (Vierkotten et al, 2007; Won et al, 2009).

4. The finding that *Inv* localizes at the TZ is surprising since a number of studies in mammalian cells and worms specifically showing that this protein is not at the TZ. I think this deserves some further discussion in the manuscript.

In the revised manuscript version, we discussed this point in connection with cell type-specific differences in terms of cilia biology as follows: “Generally, cell type-specific differences seem to be of great importance in cilia biology. For example, we detected Invs at the TZ of WT MEFs (Fig 1C and Fig EV1C). In former studies, Invs was described as a protein which is present at the BB and/or within the cilium in IMCD3 cells, in mouse renal epithelial cells and in ciliated neurons of *C. elegans* (Sang et al, 2011; Shiba et al, 2009; Warburton-Pitt et al, 2012). Thus, the ciliary localisation of Invs does not only differ between different organisms but also between different cell types within the same organism.” (p. 19 and 20).

5. References are missing from the second last sentence on Page 14.

We apologise for the oversight and inserted the missing reference (Li et al, 2016) behind the sentence “Consistent with the data from *Drosophila*, Cep290 plays a crucial role in the assembly of the TZ in *C. elegans* where it is essential for the assembly of the Mks/B9 module but not the Nphp modules at the TZ” (p. 18).

6. On page 15, the authors discuss why MKS/B9 levels do not change in *RPGRIP1L*^{-/-} MEF/kidney cilia, despite a reduction in CEP290 TZ signals. To test some of the reasons they provide for this observation, one could profile the TZ levels of MKS/B9 proteins against the levels of CEP290 *RPGRIP1L*^{-/-} cilia costained for both proteins? For example, in more extreme examples where CEP290 levels are very low (80-90% reduced), do you still see normal levels of MKS/B9 proteins? This question raised by Referee #1 is one of the open key questions within our manuscript. To get a more defined condition than to look for very low levels of Cep290 in *Rpgrip11*^{-/-} MEFs and embryonic kidneys, we generated *Cep290*^{-/-} NIH3T3 cells (immortal MEF cell line; Appendix Figs S3, S7, S9, S10C and F and Appendix Tables S1 and S2) and quantified not only the Mks/B9 module members Mks1, Tmem67, Tctn1, Tctn2 and Tctn3 (Fig EV3F-J) but also *Rpgrip11* and the Nphp module members Nphp1, Invs and Nphp4 in these cells (Fig 3C-F). In this way, we were able to demonstrate that the TZ amount of the Mks/B9 module components does not depend on the presence of Cep290 at the ciliary TZ in NIH3T3 cells (Fig EV3F-J).

7. N-values not provided for data in FigS4

N-values for this data were provided in the Figure legend of Fig 7B and C in the revised manuscript (p. 50).

8. In Fig. S3, the *ARL13* signal looks like it is in distal regions of *RPGRIP1L* $-/-$ cilia. Is this a representative image, and if so, is there any significance to this observation?

No, the *Arl13b* signal is present along the whole cilium but it is reduced in its amount in all analysed *Rpgrip11*^{-/-} cell types (MEFs, mouse embryonic kidneys, limb bud cells). We apologise for the misleading image choice in the former Fig S3C (*Arl13b* in cilia of *Rpgrip11*^{-/-} limb buds) and replaced the mentioned image by a representative one (Fig EV4E in the revised manuscript).

9. Although the IF images indicate that *RPGRIP1L* and *RPGRIP1* are expressed in all examined cells, some RNA analysis (q-PCR or similar) to support this would be useful.

The expression of *Rpgrip11* in MEFs has been investigated in a previous study (Merkestein et al, 2015). To show that *Rpgrip1* and *Rpgrip11* are present in the analysed cells and tissues, we performed Western blot studies for *Rpgrip1* in MEFs and for both proteins in wild-type limb buds (E12.5), mouse embryonic kidneys (E18.5) and HEK293 cells. By doing so, we were able to show that not only the genes are expressed but also the proteins are made by these genes in the analysed cells and tissues (Fig 4A and B, Fig 5F-H and Appendix Fig S12A and B). Furthermore, our rescue experiments clearly demonstrate the importance of *Rpgrip11* function in MEFs and of both proteins in HEK293 cells (Appendix Fig S1C and Appendix Figure S12G, J and K).

10. Details on the antigens used in FigS5 are missing.

We provided the details on the antigens in the Material and Methods section (p. 26).

Referee #2:

In this manuscript Wiegering et al. investigate the role of Rpgrip11, Nphp4 and Rpgrip1 in regulating the localization of ciliary transition zone proteins. Primary cilia are sensory organelles present on many cell types. Mutations of genes encoding for ciliary proteins can lead to a variety of diseases referred to as ciliopathies. The most prominent and frequent form is Cystic Kidney Disease, while the actual number of different ciliopathies is still increasing. All ciliopathies lack any causative therapy. Therefore, a detailed understanding of cilia function and especially the function of ciliopathy genes is urgently required. The at hand study focuses on the transition zone (TZ), which is localized between the basal body and the ciliary shaft. Here, the entry and exit in cilia is regulated and many disease genes encode to TZ proteins. The authors primarily focus on MKS5 alias Rpgrip11 or NPHP8 and aim to confirm data from C. elegans on the important role of Rpgrip11 at the

TZ. The authors show, that in murine tissues and cultured cells (and in human cells) Rpgrip11 regulates the localization of Nphp4, Nphp2 (invs), Nphp1, Nphp6 (Cep290) and other TZ players, to different extents in different cell types/tissues. Here, Rpgrip1 as synergistic partner of Rpgrip11 might be important.

In summary, this is an interesting study on an interesting cell biological and biomedical topic. The study uses extensive ciliary IF imaging and is to this end very carefully performed. However, the study remains primarily descriptive and does not gain many new mechanistic insights into the regulation of the TZ proteins that are investigated. It does not really address the question of how Rpgrip1L and Rpgrip1 actually do ensure TZ assembly. On the plus side, however, most observation were made in vivo using the appropriate genetic mouse models or at least Mef's derived from those. Although I really like this systematic and thoroughly performed imaging study, I have a number of concerns that should be addressed by the authors:

1) *Key finding of Fig. 1 and Fig. 2 is the reduction of Nphp1/4 or Nphp1/Invs at the TZ in Rpgrip11^{-/-} or in NPHP4^{-/-} MEFs, respectively. This is not entirely convincing. Here, the authors should explain more detailed the provenience of the MEFs. Are they derived from littermates sharing exactly the same genetic background? Have the results been confirmed in independent MEF cell lines to exclude any (clonal) cell culture artefacts? To ultimately prove the specificity of their*

findings Rpgrip11 or Nphp4 should be re-expressed in the respective KO MEFs. An even partial rescue in these easy experiments would underline the specificity of these data.

We used MEFs as the mouse in vitro model and they were obtained directly from mouse embryos. Thus, MEFs are not a cell line but primary cells which cannot be cultured for a long time. In comparison to cell lines, this fact limits clonal cell culture artefacts per se. The MEFs we used in this study share the same genetic background but not all of them are littermates because this is impossible. For example, *Rpgrip11*^{+/+} female mice give birth to 9 embryos on average. Considering the Mendelian ratio, it is impossible to get 3-5 wild-type and 3-5 *Rpgrip11*-negative mouse embryos out of one litter. As Referee #2 recommended, we proved our results by re-expressing full-length *Rpgrip11* and full-length *Nphp4* in the respective KO MEFs. In case of *Rpgrip11*, we used the human full-length protein because the murine full-length protein cannot be stably expressed for unknown reason. By re-expressing these proteins and by quantifying the protein amount of *Nphp1* at the TZ, we are able to show that the effects on the *Nphp1* amount in the respective KO MEFs are a consequence of the respective gene inactivation (Appendix Fig S1A-C and Appendix Fig S2F-H).

2) The authors use Crispr/Cas9 to generate cell lines lacking expression of Nphp1 and Inversin (Fig. 2 J,K). The authors should provide the genetic information of the cell lines presented. What is the nature of the on-target mutation? How did they exclude off-target effects? Either additional clones based on different sgRNAs could be used or rescue experiments should be performed to demonstrate specificity. Moreover, the knockout should be proven by western blot and not only by IF stainings of the GOI at the TZ.

We have added more information about the cell lines (genotype analysis, nature of the on-target mutation, off-target analyses) which are presented in our study and which were generated by using the CRISPR/Cas9 system (Appendix Figs S3, S6, S8, S9 and Appendix Tables S1 and S2). Moreover, we included the investigations of an additional *Invs*^{-/-} cell clone into the report (Fig 3B, Appendix Fig S10B and E) and performed rescue experiments which verify the specificity of our KO NIH3T3 cells and the respective results (Appendix Fig S10G-L). Furthermore, Western blot studies for all KO cells which were generated by using the CRISPR/Cas9 system have been added to the study and confirm the absence of the respective proteins (Appendix Fig S10A-C).

3) In Fig. 6 the authors switch to HEK cells as they state "to get an idea if these data obtained from mice are transferable to humans". There are indeed a view reports on primary cilia in HEK cells. However, HEK cells are not an ideal model to study cilia biology or human diseases. Therefore, the authors might rather use more differentiated ciliated human cell lines (e.g. RPE (retina), HK2 (kidney), ...).

Although the use of HEK293 cells can provide valuable insights into the biology of cilia since they are "likely to process nascent proteins in a physiologically relevant fashion" (Dawe et al, 2007), scientists in the cilia research field do not like to work with HEK293 cells because the handling of them is more difficult than the handling of, for example, hTERT-RPE1 cells which are commonly used. Since the mouse embryonic kidney is one of our in vivo models in our study, HEK293 cells are the ideal model to study whether the effects we observed in mice might be transferable to humans. In contrast to HEK293 cells, RPE1 cells are derived from another organ and HK2 cells were originally isolated from human kidneys but not from embryonic human kidneys.

4) As mentioned above for the Nih3t3 Crispr/Cas9 cells, the knockout RPGRIP1L and RPGRIP1 in HEK cells must be proven by western blot. What is the genotype of the cell clones? Additional clones should be added and/or re-expression experiments performed to show specificity.

As also done for the NIH3T3 cell clones, we have added more information about the HEK293 cell clones (genotype analysis, nature of the on-target mutation, off-target analyses) which are presented in our study and which were generated by using the CRISPR/Cas9 system (Appendix Figs S3, S4, S5, S9 and Appendix Tables S1 and S2). In addition, we have proven the inactivation of *RPGRIP1L* and *RPGRIP1* in these HEK293 cell clones by Western blot analyses (Appendix Fig S12A and B). Furthermore, we re-expressed *Rpgrip11* in the *RPGRIP1L*^{-/-} and the *RPGRIP1L*^{-/-}; *RPGRIP1*^{-/-} HEK293 clones (Appendix Fig S12E, F, H and I). Unfortunately, we are not able to stably express full-length *Rpgrip11*. Although we did not re-express *Rpgrip11*, we confirmed the specificity of both the *RPGRIP1L*^{-/-} and the *RPGRIP1L*^{-/-}; *RPGRIP1*^{-/-} HEK293 clones by re-expressing *Rpgrip11* and quantifying *Nphp1*. In *RPGRIP1L*^{-/-} HEK293 cells, the TZ amount of *Nphp1* was completely restored after the re-expression of *Rpgrip11* (Appendix Fig 12G). However, in *RPGRIP1L*^{-/-}; *RPGRIP1*^{-/-} HEK293 the TZ amount of *Nphp1* was significantly increased but not restored (Appendix Fig 12J and K) reflecting the absence of *Rpgrip11*.

5) Interestingly, heterozygous *Rpgrip1l* mice combined with the *Rpgrip1 nmf247* allele develop polycystic kidneys underlining their synergism *in vivo*. I feel that this important phenotypic finding should not be "hidden" in Fig S4 but presented in more detail e.g. in Fig. 5.

We have shifted this finding from Fig S4 to Fig 7B and, in addition, we performed a cystic index analysis to provide a quantification of the defect observed in *Rpgrip1l^{-/-}; Rpgrip1^{nmf247/nmf247}* embryonic kidneys (Fig 7C).

6) Most of the data presented in this study demonstrates quantitatively altered localization of *Nphp* proteins at the TZ. This raises the question whether the loss of the *Rpgrip1l/Rpgrip1* alters the architecture and ultrastructure of the transition zone? This could be studied by some more super resolution imaging (such as 3D-SIM) but ultimately might require electron microscopic imaging of the TZ in the *Rpgrip1* mutant cells.

As we have noted, 3D-SIM is only restrictedly suitable for use in investigating the ultrastructure of the transition zone. In this case, transmission electron microscopy (TEM) is the method of choice. Consequently, TEM was performed on the ciliary TZs of *Rpgrip1l^{-/-}* mouse limb buds and embryonic kidneys. The electron micrographs of both revealed that almost all Y-links were barely or not at all detectable and some of the axoneme microtubules do not exist as doublets but as singlets in the absence of *Rpgrip1l* (Fig EV4A and B). We also tried to examine the ultrastructure of the TZ in cilia of *Rpgrip1l^{-/-}; Rpgrip1^{nmf247/nmf247}*-combined mutant embryonic kidneys but we were not able to identify any TZ in the four investigated *Rpgrip1l^{-/-}; Rpgrip1^{nmf247/nmf247}* mouse embryonic kidneys. Neither we found Y-links nor any clearly detectable ciliary pocket region indicating that the ultrastructure of the TZ might be much more affected than the ultrastructure in *Rpgrip1l^{-/-}* kidneys.

Minor points:

- pg 19, methods: Regarding the *Nphp4* mice: Why (and how) was cre recombination done in ES cells?

Considering the ubiquitous expression of *Nphp4* and the major importance of primary cilia during development, we chose the strategy of the Cre-LoxP system to generate the *Nphp4* mice model. *Nphp4^{-/-}* mice were generated at the Mouse Clinical Institute – Institut Clinique de la Souris (<http://www.mci.u-strasbg.fr>; Illkirch, France). *Nphp4^{flox/flox}* mice were obtained by DNA recombination, inserting LoxP sites flanking the 5' and 3' regions of the exons 14-16 in *Nphp4* gene in embryonic stem cells line 129/Sv using a *PGK-neo* cassette as a selectable marker. These mice were mated with a Cre transgenic mouse line to obtain *Nphp4^{-/-}* mice with genomic deletion of the exons 14-16. Genotypes were determined with PCR amplification of total genomic DNA, using 2 pairs of primers: upper primer in intron 13: 5'-CTG AGA CAG GAC AGG AAC GTG ATG C-3' and lower in intron 13/exon 14 : 5'-CTG CTC TCT AAT ACA GAT TCC TGC C-3' (WT strand) or in intron 17 : 5'-TCA CAG TCA CAG TGA CTC CAA GTC C-3' (deleted strand).

The genomic deletion of exon 14 - exon 16 of the *Nphp4* gene is predicted to lead to frame-shift. Analysis of the transcripts by qPCR in MEF cells revealed a 30% reduction of *Nphp4* mRNA in *Nphp4^{-/-}* MEFs compared to *Nphp4^{+/+}* MEFs (Appendix Fig S2B). RT-PCR analysis using primers in exon 10 and exon 17 follow by SANGER sequencing detect one main abnormal *Nphp4* mRNA transcript lacking exons 12-16 (in-frame deletion; Appendix Fig S2C). This transcript is predicted to result in protein lacking AA478 to 714 (Appendix Fig S2D) encompassing part of the C2 domain of NPHP4 (AA 666-784) known to interact with RPGRIP1, RPGRIP1L (Arts et al, 2007) and KIF13B (Schou et al, 2017). Importantly, we could not detect any *Nphp4* signal in our Western blot analysis revealing that the predicted truncated protein is unstable (Appendix Fig S2E and Appendix Figure S14). Re-expression of *Nphp4* in *Nphp4^{-/-}* MEFs resulted in a restored amount of *Nphp1* at the TZ (Appendix Fig S2F-H). All these informations and data were inserted into the revised manuscript (Appendix Fig S2, Appendix Fig S14 and "Materials and Methods" on p. 25).

- The legend of Fig. 5 L does not mention the correct genotypes "(L) Illustration of several defect frequencies in WT, *Rpgrip1nmf247/nmf247* and *Rpgrip1nmf247/nmf247; Rpgrip1nmf247/nmf247* mouse embryos".

We thank Referee #2 for mentioning our mistake. The figure legend (Fig 7A in the revised manuscript) was corrected.

- The authors should provide more details and comments on the image acquisition and quantification of TZ proteins.

We have added more information about the acquisition of the images and the quantification of the TZ proteins in the Material and Methods section (p. 28, 30 and 31). In terms of the fluorescence intensity quantification, we provided more references (p. 30) to show that the method we used is commonly used in this way. Furthermore, we added the information to the fluorescence quantification of the newly performed 3D reconstruction of the 3D-SIM z-stacks.

Referee #3:

The manuscript by Wiegering et al describes extensive in vitro and in vivo cell biology studies that delineate the relative roles of the related proteins RPGRIP1L and RPGRIP1 in maintaining the ciliary transition zone. The important findings from this study are that RPGRIP1L and RPGRIP1 act synergistically to maintain the correct localization of transition zone proteins, and that RPGRIP1L contributes to this process in a cell-type specific manner.

The authors base these conclusions on quantitative immunofluorescence studies using an extensive panel of mouse embryonic fibroblasts, kidneys and limb buds derived from Rpgrip1l, Rpgrip1 and Nphp4 knock-out mice. Double knock-out lines in HEK293 cells have also been generated and characterized. However, in all of these experiments, the "total pixel intensity" method of quantitation is unclear to me. In each of the bar graphs, I assume that the black bars are intensity values from wild-type cells normalized to 100% and grey bars are mutant cells. Comparisons using total pixel intensity could then be easily confounded by technical issues and variations in staining efficiency. For me, a more robust method would be a measure of co-localization within each experiment using a 3D reconstruction from a confocal z-stack. This is reasonably straight-forward in software such as Volocity, and possible with more effort in FIJI.

The method of quantification we used in this manuscript was performed by several other studies in this way. To increase the transparency of the method, we inserted more details into the "Quantifications" section in "Materials and Methods" (p. 30 and 31). To verify that this is an established quantification method, we added several references in the same section. In the revised manuscript, we explain that the black bars (intensity values of the WT) are normalised to 100% (see Figure legends) to avoid any misunderstandings. To test the quality and precision of the used method of quantification, we made a 3D reconstruction of our z-stacks which were obtained via 3D-SIM to get a higher resolution than in the case of simple confocal images. We performed this experiment exemplarily for the Cep290 staining at the ciliary TZ in WT and *Rpgrip1l*^{-/-} MEFs. Then, we quantified the Cep290 staining as described in the "Materials and Methods" section in the revised manuscript: "Intensity quantification of the Cep290 fluorescence signal which was obtained by using 3D-SIM was performed as follows: Surface reconstructions of Cep290 were performed using Imaris 8.3 (Bitplane). 3D-SIM images of WT and *Rpgrip1l*^{-/-} MEFs were analysed in batch mode applying a manual threshold of 13,000 and a surface area detail of 50 nm. For each image, we manually selected the surface of Cep290 between the BB (γ -tubulin) and the ciliary axoneme (acetylated α -tubulin) to measure the summarised intensity. If necessary two signals were separated manually using the cut surface function of Imaris. For images where no surface were detected in the TZ measurements were set to 0." (p. 31). By doing so, we confirmed the reduced amount of Cep290 at the TZ of *Rpgrip1l*^{-/-} MEFs (Fig 1E and Movie EV1 and EV2) validating the precision of our measurement procedure.

A more subtle effect could be possible from differences in cell cycle progression between wild-type and mutant cell-lines, or differences in the relative rates of either ciliogenesis or transition zone assembly. To exclude the latter possibility, live cell imaging could be done in the MEF or knock-out NIH3T3 or HEK293 cell-lines. The authors' work suggests that CEP290 localization would be a good one to test.

We agree with Referee #3 in the point that it cannot be excluded that differences in cell cycle progression, ciliogenesis or TZ assembly between the WT and the mutants affect the amount of TZ proteins. However, it is impossible to correlate these parameters via live cell imaging. To quantify fluorescence intensities in a live cell imaging study, you have to transfect the cells with plasmids encoding for TZ proteins fused to a fluorescence marker. Unfortunately, the expression of these fusion proteins varies after transfection depending on e.g. the transfection efficiency. Consequently, a quantification of fluorescence signals which are based on transfection will be inaccurate and provide no reliable results. Like it was done in other studies (Garcia-Gonzalo et al, 2011; Garcia-Gonzalo et al, 2015; Gerhardt et al, 2015; Roberson et al, 2015; Struchtrup et al, 2018; Yee et al,

2015), we quantified the amount of ciliary proteins in a high sample size (see Figure legends for the biological n and the number of cilia which were used in the individual measurements). In this way, we minimise sources of error because even in serum-starved cells, not all cilia are in the same stage of ciliogenesis. While some cilia are formed, others are degraded. This is even the case in the WT cells. In our opinion, the high sample size method is best possibility to generate reliable data.

The work also suggests that RPGRIP1 is able to compensate, at least partially, for RPGRIP1L in MEFs (Figure 5). Since the phenotypic effects of human RPGRIP1 mutations are limited to the photoreceptors (causing Leber congenital amaurosis), it would therefore also be interesting to test the extent of RPGRIP1L/RPGRIP1 synergy in the photoreceptor connecting cilium. One might predict that RPGRIP1 would not be able to compensate for RPGRIP1L because the retinal-specific function of RPGRIP1.

Rpgrip11^{-/-} mouse embryos display severe eye defects, while Rpgrip11^{+/-} mouse embryos do not show any abnormalities in eye development (Vierkotten et al, 2007). Rpgrip1^{+/nmf247} mice do not exhibit any eye defects, while Rpgrip1^{nmf247/nmf247} mice suffer from defective eye development from P12-14 on (Won et al, 2009). To evaluate whether there is a synergistic effect of Rpgrip11 and Rpgrip1 on eye development, we analysed the eye phenotype of Rpgrip11^{+/-}; Rpgrip1^{+nmf247}. In contrast to Rpgrip1^{nmf247/nmf247} eyes, Rpgrip11^{+/-}; Rpgrip1^{+nmf247} eyes at P28 display no defects (see images below). Since we did not use the eye as an in vivo model in the manuscript, we did not present these data in the revised version.

RPE: retinal pigment epithelium; NL: nuclear layer; GC: ganglion cells.

However, what I miss in these experiments is a more detailed consideration of the possible mechanistic or structural basis of RPGRIP1L/RPGRIP1 synergy. RPGRIP1L and RPGRIP1 appear to have different localizations to the transition zone (Figure 2A) and basal body (Figure 4C-E), respectively, at least in MEFs. (This is also shown in Figure 4B). Could super resolution microscopy allow better characterization of this localization and ask if other transition zone proteins are mis-localized? For example, CEP290 appears to be mis-localized in Nphp4 knock-out MEFs compared to wild-type (Figure 2D).

We have already investigated the localisation of Rpgrip11 and Rpgrip1 in relation to each other by using super-resolution microscopy (3D-SIM) and detected these proteins close to each other (Fig 5C in the revised manuscript). In this context, Rpgrip11 is present between Rpgrip1 and the axoneme. We performed additional stainings in which we stained the basal body with γ -tubulin and Rpgrip1 or Rpgrip11 (Fig 5B and Fig EV1E in the revised manuscript). These immunofluorescence studies confirmed the localisation of Rpgrip1 at the basal body and the localisation of Rpgrip11 in the transition zone. Moreover, we performed an in situ proximity ligation assay (in situ PLA) to estimate the distance between the two proteins. Since a PLA signal was detected at the ciliary base, the distance between Rpgrip11 and Rpgrip1 is smaller than 40 nm.

We agree with Reviewer #3 in the point that Figure 2D could be interpreted as a mis-localisation of Cep290 in *Nphp4*^{-/-} MEFs. Thus, we analysed the localisation of Cep290 in these MEFs by using 3D-SIM. By doing so, we could not find a mis-localisation of Cep290 in the absence of *Nphp4* (Fig EV1D). Consequently, we replaced the images in Figure 2D by representative images. Moreover, we have proven the localisation of other TZ proteins by using 3D-SIM (Fig EV1) and could not

observe any mis-localisation. These data indicate that the absence of all analysed transition zone proteins does not alter the localisation of other TZ proteins but only their transition zone amount.

Finally, the phrase "mice and men" is used a couple of times, and although I also appreciate the poetry of Robert Burns, I think that it is inappropriate in this context because it could be misconstrued as sexist.

We revised these phrases. Of course it was not our intention to write any sexist phrases. Such an intention would be inexcusable in general and especially also from the point of view that 62.5% of the authors (including the first author) are female in this case.

References:

Arts H, Doherty D, van Beersum S, Parisi M, Letteboer S, Gorden N, Peters T, Märker T, Voesenek K, Kartono A, Ozyurek H, Farin F, Kroes H, Wolfrum U, Brunner H, Cremers F, Glass I, Knoers N, Roepman R (2007) Mutations in the gene encoding the basal body protein RPGRIP1L, a nephrocystin-4 interactor, cause Joubert syndrome. *Nat Genet* **39**: 882-888.

Dawe H, Smith U, Cullinane A, Gerrelli D, Cox P, Badano J, Blair-Reid S, Sriram N, Katsanis N, Attié-Bitach T, Afford S, Copp A, Kelly D, Gull K, Johnson C (2007) The Meckel-Gruber Syndrome proteins MKS1 and meckelin interact and are required for primary cilium formation. *Hum Mol Genet* **16**: 173-186.

Garcia-Gonzalo F, Corbit K, Sirerol-Piquer M, Ramaswami G, Otto E, Noriega T, Seol A, Robinson J, Bennett C, Josifova D, García-Verdugo J, Katsanis N, Hildebrandt F, Reiter J (2011) A transition zone complex regulates mammalian ciliogenesis and ciliary membrane composition. *Nat Genet* **43**: 776-784.

Garcia-Gonzalo F, Phua S, Roberson E, Garcia Gr, Abedin M, Schurmans S, Inoue T, Reiter J (2015) Phosphoinositides Regulate Ciliary Protein Trafficking to Modulate Hedgehog Signaling. *Dev Cell* **34**: 400-409.

Gerhardt C, Lier J, Burmühl S, Struchtrup A, Deutschmann K, Vetter M, Leu T, Reeg S, Grune T, Rütger U (2015) The transition zone protein Rpgrip11 regulates proteasomal activity at the primary cilium. *J Cell Biol* **210**: 115-133.

Li C, Jensen V, Park K, Kennedy J, Garcia-Gonzalo F, Romani M, De Mori R, Bruel A, Gaillard D, Doray B, Lopez E, Rivière J, Faivre L, Thauvin-Robinet C, Reiter J, Blacque O, Valente E, Leroux M (2016) MKS5 and CEP290 Dependent Assembly Pathway of the Ciliary Transition Zone. *PLoS Biol* **14**: e1002416.

Merkestein M, Laber S, McMurray F, Andrew D, Sachse G, Sanderson J, Li M, Usher S, Sellayah D, Ashcroft F, Cox R (2015) FTO influences adipogenesis by regulating mitotic clonal expansion. *Nat Commun* **6**: 6792.

Roberson E, Dowdle W, Ozanturk A, Garcia-Gonzalo F, Li C, Halbritter J, Elkhartoufi N, Porath J, Cope H, Ashley-Koch A, Gregory S, Thomas S, Sayer J, Saunier S, Otto E, Katsanis N, Davis E, Attié-Bitach T, Hildebrandt F, Leroux M, Reiter J (2015) TMEM231, mutated in orofacioidigital and Meckel syndromes, organizes the ciliary transition zone. *J Cell Biol* **209**: 129-142.

Sang L, Miller J, Corbit K, Giles R, Brauer M, Otto E, Baye L, Wen X, Scales S, Kwong M, Huntzicker E, Sfakianos M, Sandoval W, Bazan J, Kulkarni P, Garcia-Gonzalo F, Seol A, O'Toole J, Held S, Reutter H, Lane W, Rafiq M, Noor A, Ansar M, Devi A, Sheffield V, Slusarski D, Vincent J, Doherty D, Hildebrandt F, Reiter J, Jackson P (2011) Mapping the NPHP-JBTS-MKS protein network reveals ciliopathy disease genes and pathways. *Cell* **145**: 513-528.

Schou K, Mogensen J, Morthorst S, Nielsen B, Aleliunaite A, Serra-Marques A, Fürstenberg N, Saunier S, Bizet A, Veland I, Akhmanova A, Christensen S, Pedersen L (2017) KIF13B establishes

a CAV1-enriched microdomain at the ciliary transition zone to promote Sonic hedgehog signalling. *Nat Commun* **8**: 14177.

Shi X, Garcia Gr, Van De Weghe J, McGorty R, Pazour G, Doherty D, Huang B, JF. R (2017) Super-resolution microscopy reveals that disruption of ciliary transition-zone architecture causes Joubert syndrome. *Nat Cell Biol* **19**: 1178-1188.

Shiba D, Yamaoka Y, Hagiwara H, Takamatsu T, Hamada H, Yokoyama T (2009) Localization of Inv in a distinctive intraciliary compartment requires the C-terminal ninein-homolog-containing region. *J Cell Sci* **122**: 44-54.

Struchtrup A, Wiegeling A, Stork B, R  ther U, Gerhardt C (2018) The ciliary protein RPGRIP1L governs autophagy independently of its proteasome-regulating function at the ciliary base in mouse embryonic fibroblasts. *Autophagy*: 1-47.

Vierkotten J, Dildrop R, Peters T, Wang B, R  ther U (2007) Ftm is a novel basal body protein of cilia involved in Shh signalling. *Development* **134**: 2569-2577.

Warburton-Pitt S, Jauregui A, Li C, Wang J, Leroux M, Barr M (2012) Ciliogenesis in *Caenorhabditis elegans* requires genetic interactions between ciliary middle segment localized NPHP-2 (inversin) and transition zone-associated proteins. *J Cell Sci* **125**: 2592-2603.

Won J, Gifford E, Smith R, Yi H, Ferreira P, Hicks W, Li T, Naggert J, Nishina P (2009) RPGRIP1 is essential for normal rod photoreceptor outer segment elaboration and morphogenesis. *Hum Mol Genet* **18**: 4329-4339.

Yee L, Garcia-Gonzalo F, Bowie R, Li C, Kennedy J, Ashrafi K, Blacque O, Leroux M, Reiter J (2015) Conserved Genetic Interactions between Ciliopathy Complexes Cooperatively Support Ciliogenesis and Ciliary Signaling. *PLoS Genet* **11**: e1005627.

2nd Editorial Decision

1 March 2018

Thank you for submitting a revised version of your manuscript. The manuscript has now been seen by the three original referees, who find that their main concerns have been addressed and are now broadly in favour of publication of the manuscript. There remain only a few minor issues that have to be dealt with before I can extend formal acceptance of the manuscript.

1. Please clarify the remaining questions on CRISPR/Cas9 knockout cell line generation and implement the minor phrasing adjustments requested by referee #2.
2. Please clarify the discrepancy between the contradictory data on the synergistic effect of Rpgrip1 and Rpgrip1l in the eye (referee #3).

Referee #1:

The authors have done a very good job in addressing my comments.

Referee #2:

In this revised version of the MS the authors have addressed almost all of my concerns.

I still do not entirely understand how the KO cell lines have been generated. The description in the

"Appendix M&M" section is not absolutely clear. Why are there some clones with not just 2, but 3 or 4 different allele? Are these in the end pooled/mixed clones? In addition, "targeted" off-target analyses by RFLP might not be the strongest way to exclude unwanted effects. However, some re-expression data has been added.

I am still not convinced of HEK293 cells as a model system (for human embr. kidney !), but this is not a major issue for the at hand story. However, I feel that the sentence "To get an idea if these data obtained from mice are transferable to humans ..." should probably be rephrased in the context of using highly artificial HEK293 cells.

Referee #3:

Thank you for the new re-submission of the manuscript by Wiegering et al. The authors have satisfied one of my major concerns about the methodology that they used to determine co-localization, particularly with a relatively new super-resolution microscopy method such as 3D-SIM, and the further details will make any independent replication an easier task. In this vein, I also appreciated the more detailed description of crispant line derivation and characterization. I also appreciated the additional efforts to exclude the possible mis-localization of CEP290, now presented in supplementary material.

I accept that the data presented in the rebuttal for *Rpgrip11^{+/-};Rpgrip1^{+/nmf247}* mice probably precludes any *RPGRIP1L/RPGRIP1* synergistic effect on eye development. However, the data in the new Figure 7 now shows the synergistic effect of the *nmf247* allele on kidney cyst formation. The same data (Figure 7A) also suggests a small effect on "eye defects". What are these defects and is the effect statistically significant? If it is, then they are, then my original suggestion to test *NPHP/TZ* protein co-localization in the photoreceptor still stands.

2nd Revision - authors' response

12 March 2018

Referee #2:

In this revised version of the MS the authors have addressed almost all of my concerns.

I still do not entirely understand how the KO cell lines have been generated. The description in the "Appendix M&M" section is not absolutely clear. Why are there some clones with not just 2, but 3 or 4 different allele? Are these in the end pooled/mixed clones? In addition, "targeted" off-target analyses by RFLP might not be the strongest way to exclude unwanted effects. However, some re-expression data has been added.

We revised the Appendix Materials and Methods section in order to clarify the points raised by Referee #2. In this context, we provided more precise information about the karyotype and why we found less alleles than the expected number (see Appendix Materials and Methods p. 4).

I am still not convinced of HEK293 cells as a model system (for human embr. kidney !), but this is not a major issue for the at hand story. However, I feel that the sentence "To get an idea if these data obtained from mice are transferable to humans ..." should probably be rephrased in the context of using highly artificial HEK293 cells.

The words "To get an idea if these data obtained from mice are transferable to humans ..." were replaced by the sentence "Moreover, we investigated the role of *Rpgrip11* and *Rpgrip1* in TZ assembly in the human cell line HEK293 (human embryonic kidney 293)." in the Results section (p. 14). Furthermore, the sentence "To evaluate whether it is possible to transfer our results from mice to humans, we generated *RPGRIP1L^{-/-}* and *RPGRIP1L^{-/-}; RPGRIP1^{-/-}* HEK293 cells (Appendix Fig S12A-K)." was deleted in the Discussion section. Instead, we wrote "Importantly, the data obtained from *RPGRIP1L^{-/-}* and *RPGRIP1L^{-/-}; RPGRIP1^{-/-}* HEK293 cells confirmed the results quantified in MEFs and murine embryonic kidneys making it very likely that the TZ assembly organisation in

humans is similar to that in mice (Fig 1A, C, D and E and Fig 4C, E, F and J and Fig 6A, C, D and H and Fig 8A-D and Fig EV3D).” (p. 22).

Referee #3:

Thank you for the new re-submission of the manuscript by Wiegeling et al. The authors have satisfied one of my major concerns about the methodology that they used to determine co-localization, particularly with a relatively new super-resolution microscopy method such as 3D-SIM, and the further details will make any independent replication an easier task. In this vein, I also appreciated the more detailed description of crispant line derivation and characterization. I also appreciated the additional efforts to exclude the possible mis-localization of CEP290, now presented in supplementary material.

I accept that the data presented in the rebuttal for $Rpgrip11^{+/-}; Rpgrip1^{+}/nmf247$ mice probably precludes any RPGRIP1L/RPGRIP1 synergistic effect on eye development. However, the data in the new Figure 7 now shows the synergistic effect of the $nmf247$ allele on kidney cyst formation. The same data (Figure 7A) also suggests a small effect on "eye defects". What are these defects and is the effect statistically significant? If it is, then they are, then my original suggestion to test NPHP/TZ protein co-localization in the photoreceptor still stands.

We agree with Referee #3 that the data we presented in the previous version of our manuscript allow the presumption that $Rpgrip11$ and $Rpgrip1$ may control proper eye development synergistically even if eye morphogenesis seems to be unaltered in $Rpgrip11^{+/-}; Rpgrip1^{+}/nmf247$ mice. Since this presumption is mainly based on the defect frequency illustration in Figure 7A, we followed the advice of Referee #3 and tested for significance by using the chi-square test. Remarkably, the difference in eye defect frequency (comprising microphthalmia and deep-set (in the “centre” of the head) eyes) between $Rpgrip11^{-/-}$ and $Rpgrip11^{-/-}; Rpgrip1^{nmf247}/nmf247$ mouse embryos is not significant. However, the ventricular septal defect frequency is significantly increased in $Rpgrip11^{-/-}; Rpgrip1^{nmf247}/nmf247$ mouse embryos. But since the main conclusion of our study is that vertebrate TZ assembling is regulated in a cell type-specific manner, the quantification of the TZ amount of all analysed proteins additionally in cardiac cilia would be a time consuming endeavour without any further statement.

Corresponding Author Name: Dr. Christoph Gerhardt

Journal Submitted to: The EMBO Journal

Manuscript Number: EMBOJ-2017-97791